# Certifying Concavity and Monotonicity in Games via Sum-of-Squares Hierarchies

**Vincent Leon**[1]    **Iosif Sakos**[2]    **Ryann Sim**[2]    **Antonios Varvitsiotis**[2,3,4]

[1]UIUC    [2]SUTD    [3]NUS CQT    [4]Archimedes/Athena RC

`leon18@illinois.edu`

`{iosif_sakos, ryann_sim, antonios}@sutd.edu.sg`

## Abstract

Concavity and its refinements underpin tractability in multiplayer games, where players independently choose actions to maximize their own payoffs which depend on other players' actions. In *concave* games, where players' strategy sets are compact and convex, and their payoffs are concave in their own actions, strong guarantees follow: Nash equilibria always exist and decentralized algorithms converge to equilibria. If the game is furthermore *monotone*, an even stronger guarantee holds: Nash equilibria are unique under strictness assumptions. Unfortunately, we show that *certifying* concavity or monotonicity is NP-hard, already for games where utilities are multivariate polynomials and compact, convex basic semialgebraic strategy sets—an expressive class that captures extensive-form games with imperfect recall. On the positive side, we develop two hierarchies of sum-of-squares programs that certify concavity and monotonicity of a given game, and each level of the hierarchies can be solved in polynomial time. We show that almost all concave/monotone games are certified at some finite level of the hierarchies. Subsequently, we introduce the classes of SOS-concave/monotone games, which globally approximate concave/monotone games, and show that for any given game we can compute the closest SOS-concave/monotone game in polynomial time. Finally, we apply our techniques to canonical examples of extensive-form games with imperfect recall.

## 1   Introduction

Game theory models settings where multiple decision-makers independently maximize personal objectives that depend on the actions of others. Formally, a game with $n$ players is modeled by assigning to each player $i$ a strategy set $\mathcal{X}_i \subset \mathbb{R}^{m_i}$ and a utility function $u_i(x_i, x_{-i})$, where $x_i \in \mathcal{X}_i$ is player $i$'s action and $x_{-i}$ denotes the actions of all other players. The interdependence of players' utilities makes analyzing the collective behavior of such systems both rich and challenging.

The canonical solution concept in game theory is the *Nash equilibrium* [42], a product distribution over strategies in which no player can unilaterally deviate to improve their utility, given the strategies of the other players. While Nash equilibria are guaranteed to exist in finite normal-form games, several key questions must be addressed in games with *continuous, infinite* action spaces: Do Nash equilibria always exist? If one exists, is it unique (thereby avoiding the equilibrium selection problem)? And crucially, can it be computed efficiently using distributed algorithms?

Extensive research has identified *concavity*, and its refinements, as key enablers in addressing these fundamental questions. In the setting of games, this entails assuming that each strategy set $\mathcal{X}_i$ is compact and convex. Furthermore, concavity of players' utilities can manifest in at least two

distinct forms. First, we have the class of *concave games*, where for each player $i$, the function $x_i \mapsto u_i(x_i, x_{-i})$ is continuous and concave for every fixed $x_{-i}$. Second, we have the more restrictive class of *monotone games*, where utility functions are smooth and the (negative) concatenated gradient map

$$x \mapsto (-\nabla_{x_1} u_1(x_1, x_{-1}), \ldots, -\nabla_{x_n} u_n(x_n, x_{-n}))$$

is a *monotone operator*. Every monotone game is concave, but the converse does not necessarily hold. Concave and monotone games were first studied in the seminal work of Rosen [49], who established that Nash equilibria always exist in these class of games, significantly extending the guarantees of classical results such as von Neumann's minimax theorem [62] for two-player zero-sum (normal-form) games, Nash's aforementioned result for finite normal-form games, and Sion's minimax theorem for two-player convex-concave games [53]. Moreover, Rosen showed that *strictly monotone games* have a *unique* Nash equilibrium. At the same time, concave/monotone games have been extensively studied due to their inherent expressibility – they have been used to model various fundamental settings in economics and optimization, including but not limited to resource allocation [50], Cournot competition [17] and robust power management [67].

Finally, concave and monotone games have also received considerable attention in the context of equilibrium computation. A substantial body of work has analyzed decentralized dynamics that achieve strong performance guarantees in concave games [17, 56, 28, 57]. Most recently, [20] established a $\mathcal{O}(\text{polylog } T)$ regret bound for uncoupled learning dynamics in general convex games, extending classical results beyond structured settings such as normal-form and extensive-form games. In monotone games, decentralized dynamics have also been shown to converge in the *last-iterate sense* to Nash equilibria [38, 8, 9, 25, 18]. Moreover, in strictly monotone games, one can guarantee last-iterate convergence to the *unique* Nash equilibrium [67, 7, 51, 54], further underscoring the computational tractability of this class.

However, despite their favorable properties, it is not clear how to efficiently verify concavity and monotonicity. For instance, establishing that $u_i$ is concave over $\mathcal{X}_i$ requires checking that the Hessian $\nabla^2_{x_i} u_i(x_i, x_{-i})$ is negative semidefinite for every $x_i \in \mathcal{X}_i$ and every $x_{-i} \in \mathcal{X}_{-i}$, an infinite family of conditions. In view of this, a fundamental computational challenge arises:

*Is it possible to efficiently verify that a game is concave or monotone?*

**Our Techniques and Contributions.** Our starting point is to demonstrate that deciding whether a game is concave or monotone is computationally hard, cf. Theorem 3.1. We establish this hardness result for the class of polynomial games [14, 30, 45] in which each player's utility is a multivariate polynomial and players' strategy sets are compact convex basic semialgebraic sets – that is, sets defined by polynomial equality and inequality constraints. This class is highly expressive, capturing for instance extensive-form games with imperfect recall [46]. Our hardness result builds on recent advances in polynomial optimization [4, 2], which show that unless $\text{P} = \text{NP}$, there is no polynomial-time (or even pseudo-polynomial-time) algorithm that can decide whether a multivariate polynomial of degree four (or any higher even degree) is globally convex. This result presents a challenge for game theorists. On one hand, concave/monotone games are expressive classes of games that capture many applications and have desirable equilibration properties. However, verifying their concavity/monotonicity is hard for the class of polynomial games over convex compact basic semialgebraic sets.

Motivated by this, we next seek to identify tractable sufficient conditions for concavity and monotonicity, as well as special classes of games for which these properties can be efficiently certified. Our approach is based on the observation that, since polynomial games are smooth, these properties can be verified via the positive semidefiniteness of the Hessian and the symmetrized Jacobian, respectively. As a concrete example, a polynomial game is concave if, for each player $i$, the (negative) Hessian of the utility function is positive semidefinite for all $x_i \in \mathcal{X}_i$ and $x_{-i} \in \mathcal{X}_{-i}$. By the variational characterization of positive semidefiniteness, this is equivalent to requiring that $p_i(x, y) := -y^\top \nabla^2_{x_i} u_i(x_i, x_{-i}) y \geq 0$, for all $x \in \times_{i=1}^n \mathcal{X}_i$ and $y \in \mathcal{B}$, where $\mathcal{B} \subset \mathbb{R}^{m_i}$ is the unit ball. Since $u_i$ is a polynomial and $\mathcal{X}$ is a closed basic semialgebraic set, the function $p_i(x, y)$ is a polynomial over a semialgebraic domain.

Although testing nonnegativity of polynomials is, in general, computationally hard [41], a powerful approach from polynomial optimization, pioneered in [45, 35], is to seek a *sum-of-squares (SOS)* decomposition that certifies nonnegativity. This idea has also been recently used to develop certifi-

cates for the global convexity of polynomials [5]. Searching for an SOS decomposition of bounded degree can be done in polynomial time via semidefinite programming.

In our setting, the application of the SOS framework leads to a hierarchy of increasingly stronger sufficient conditions for certifying concavity or monotonicity, each of which can be checked in polynomial time via semidefinite programming. At the $\ell$-th level of the hierarchy, we check whether $p_i(x, y)$ admits a degree-$\ell$ SOS decomposition over $\mathcal{X} \times \mathcal{B}$. While the SOS framework does not eliminate the inherent hardness of the problem, it offers a practical trade-off: by relaxing the problem into a sequence of SDPs, one obtains a hierarchy of increasingly tight sufficient conditions with provable convergence in the limit. The main limitation is that the size of the resulting SDPs grows with the level of the hierarchy.

Leveraging these ideas, our main contributions are summarized below:

- We construct a hierarchy of optimization problems that provide increasingly strong certificates of monotonicity/concavity for polynomial games over compact, convex basic semialgebraic sets (cf. Theorem 3.2). Furthermore, each level of the hierarchy can be solved in polynomial time via semidefinite programming.

- We show that for every strictly monotone/strictly concave game, a certificate is always found at a *finite* level of the hierarchy (cf. Statement 4 in Theorem 3.2). More importantly, we show that for *almost all* monotone/concave games, such a certificate can be obtained at some finite level of the hierarchy (cf. Theorem 3.3).

- We define subclasses of monotone/concave polynomial games over compact, convex basic semialgebraic sets, called $\ell$-SOS-monotone (resp. $\ell$-SOS-concave) games, for which monotonicity (resp. concavity) can be certified by the $\ell$-th level of the hierarchy (cf. Definition 4.1). We show that this class of games globally approximates the class of monotone (resp. concave) games, and importantly, given any polynomial game, the closest $\ell$-SOS-monotone (resp. $\ell$-SOS-concave) game can be computed by solving a *single* SDP (cf. Theorem 4.3).

- We apply our proposed methods to several canonical and new examples of extensive-form games with imperfect recall (cf. Section 6). We show examples of how our hierarchies can be used to verify monotonicity/concavity in these games, as well as to find the closest $\ell$-SOS-monotone (resp. $\ell$-SOS-concave) game with respect to an appropriate norm.

## 2 Preliminaries

### 2.1 Polynomial Games over Semialgebraic Sets

We consider an $n$-player continuous game denoted by $\mathscr{G} = \mathscr{G}(\llbracket n \rrbracket, \mathcal{X}, u)$. For each player $i \in \llbracket n \rrbracket$, we denote their set of actions by $\mathcal{X}_i \subseteq \mathbb{R}^{m_i}$ and their payoff function by $u_i \colon \mathcal{X} \to \mathbb{R}$, where the set of joint actions $\mathcal{X} \overset{\text{def}}{=} \mathcal{X}_1 \times \cdots \times \mathcal{X}_n$ is a *compact, convex set*. Each player $i$ selects an action $x_i \in \mathcal{X}_i$. We denote by $x \overset{\text{def}}{=} (x_1, \ldots, x_n)$ the joint action profile of all players, and by $\mathcal{X} \overset{\text{def}}{=} \mathcal{X}_1 \times \cdots \times \mathcal{X}_n \subseteq \mathbb{R}^m$ their joint action space, where $m \overset{\text{def}}{=} m_1 + \cdots + m_n$. We also denote by $u \overset{\text{def}}{=} (u_1, \ldots, u_n)$ the ensemble of the players' payoff functions.

In this work, we focus on games $\mathscr{G}$ where $u_1, \ldots, u_n$ are *polynomial* functions, and $\mathcal{X}$ is a basic semialgebraic set. In particular, we assume that

$$\mathcal{X} \equiv \left\{ x \in \mathbb{R}^{m_1} \times \cdots \times \mathbb{R}^{m_n} \;\middle|\; \begin{array}{ll} g_j(x) \geq 0, & j \in \llbracket m_g \rrbracket, \\ h_j(x) = 0, & j \in \llbracket m_h \rrbracket \end{array} \right\}, \tag{1}$$

where $g_1, \ldots, g_{m_g}, h_1, \ldots, h_{m_h} \in \mathbb{R}[x]$.

We refer to $d = \max\{\deg(u_1), \ldots, \deg(u_n), \deg(g_1), \ldots, \deg(g_{m_g}), \deg(h_1), \ldots, \deg(h_{m_h})\}$ as the degree of the game. For each $n, d \in \mathbb{N}$, we use $\mathcal{G}_{(n,d)}$ to denote the set of $n$-player, $d$-degree *polynomial* games over $\mathcal{X}$.

$\mathcal{G}_{(n,d)}$ is isomorphic to $\mathbb{R}^M$, where $M \overset{\text{def}}{=} n \cdot \binom{m+d}{d}$. In particular, we define the isomorphism

$$\mathscr{G} \mapsto \left( \mathrm{vec}(u_1), \ldots, \mathrm{vec}(u_n) \right)^{\mathsf{T}}, \qquad \forall \mathscr{G} \in \mathcal{G}_{(n,d)}, \tag{2}$$

where $\mathrm{vec}(u_i)$ is the coefficient vector of $u_i$ for each $i \in [\![n]\!]$. Throughout the paper, we also consider the topology on $\mathcal{G}_{(n,d)}$ induced by the norm

$$\|\mathscr{G}\| = \max_{i \in [\![n]\!]} \|\mathrm{vec}(u_i)\|_\infty. \tag{3}$$

When necessary, we use the convention $x = (x_i, x_{-i})$ to distinguish the action $x_i$ of player $i$ in a joint action $x \in \mathcal{X}$ from the actions of the rest of the players. In a similar vein, we use $\mathcal{X}_{-i}$ to denote the joint action space of all players except player $i$.

A fundamental equilibrium concept in game theory is the Nash equilibrium (NE) [42], which are strategy profiles from which players have no incentive to unilaterally deviate. Concretely, a joint action profile $x^* \in \mathcal{X}$ is a NE of a game $\mathscr{G}$ if

$$u_i(x^*) \geq u_i(x_i, x^*_{-i}), \qquad \forall x_i \in \mathcal{X}_i, \quad i \in [\![n]\!]. \tag{4}$$

## 2.2 Sum-of-Squares Optimization

Given a closed basic semialgebraic set $\mathcal{X}$ as in (1), the quadratic module $Q(\mathcal{X})$ of $\mathcal{X}$ is a set of functions defined as

$$Q(\mathcal{X}) \overset{\mathrm{def}}{=} \left\{ \sigma_0 + \sum_{j=1}^{m_g} g_j \sigma_j + \sum_{j=1}^{m_h} h_j p_j \; \middle| \; \begin{array}{l} \sigma_0, \ldots, \sigma_{m_g} \in \Sigma[x], \\ p_1, \ldots, p_{m_h} \in \mathbb{R}[x] \end{array} \right\}, \tag{5}$$

where $\Sigma[x] \subset \mathbb{R}[x]$ is the set of sum-of-squares (SOS) polynomials on variables $x$, i.e., the set of all polynomials of the form

$$\sigma(x) = \sum_{k=1}^{K} q_k^2(x), \qquad \forall x \in \mathbb{R}^m, \qquad \text{where} \qquad q_1, \ldots, q_K \in \mathbb{R}[x]. \tag{6}$$

Furthermore, for all $d \geq 0$, we define $Q_d(\mathcal{X})$ as the restriction of $Q(\mathcal{X})$ to Putinar-type decompositions of degree at most $2d$ given by $\deg(\sigma_0), \ldots, \deg(\sigma_{m_g}), \deg(p_1), \ldots, \deg(p_{m_h}) \leq 2d$.

As part of the analysis in Section 3.1, we require that the quadratic module $Q(\mathcal{X})$ is Archimedean, a property formally given for completeness in the following definition.

**Definition 2.1.** A quadratic module $Q(\mathcal{X})$ is called Archimedean if there exists $N \in \mathbb{N}$ such that

$$N - \sum_{i=1}^{m} x_i^2 \in Q(\mathcal{X}). \tag{7}$$

## 2.3 Concave & Monotone Games

In this section, we introduce two important subclasses of continuous games, concave games and monotone games, both defined in [49]. These classes are particularly significant due to their implications for the existence and uniqueness of Nash equilibria.

**Definition 2.2** (Concave Games). A game $\mathscr{G}$ is concave if, for all players $i \in [\![n]\!]$, the function $x_i \mapsto u_i(x_i, x_{-i})$ is concave for every fixed $x_{-i} \in \mathcal{X}_{-i}$. Furthermore, if $\mathscr{G}$ is polynomial then it is concave if and only if the Hessian matrices of the payoff functions $u_1, \ldots, u_n$ with respect to $x_1$, $\ldots, x_n$, respectively, are negative semidefinite, i.e.,

$$\mathbf{H}_{u_i}(x) \overset{\mathrm{def}}{=} \nabla_{x_i}^2 u_i(x) \preceq 0, \qquad \forall x \in \mathcal{X}, \quad i \in [\![n]\!]. \tag{8}$$

Rosen [49] proved that a Nash equilibrium exists in every concave game, thereby extending Nash's equilibrium existence result to a broad class of continuous games. He also identified an important subclass of concave games with additional structural properties, which are now typically referred to as monotone games [38].

**Definition 2.3** (Monotone Games). A game $\mathscr{G}$ is monotone if the negative of its concatenated gradient mapping, referred to by Rosen as the pseudogradient,

$$v(x) \overset{\mathrm{def}}{=} \left( \nabla_{x_1}^\mathsf{T} u_1(x), \ldots, \nabla_{x_n}^\mathsf{T} u_n(x) \right)^\mathsf{T} \tag{9}$$

is a monotone operator on $\mathcal{X}$, i.e.,

$$\langle v(x) - v(x'), \, x - x' \rangle \leq 0, \qquad \forall x, \, x' \in \mathcal{X}. \tag{10}$$

Furthermore, if $\mathscr{G}$ is polynomial, it is well-known that (cf. [48, Proposition 12.3]) it is monotone if and only if the symmetrized Jacobian matrix with respect to $v(x)$ is negative semidefinite, i.e.,

$$\mathbf{SJ}(x) \stackrel{\text{def}}{=} \frac{1}{2}\big(\mathbf{J}(x) + \mathbf{J}(x)^{\mathsf{T}}\big) \preceq 0, \qquad \forall x \in \mathcal{X}, \tag{11}$$

where, for all $x \in \mathcal{X}$, $\mathbf{J}(x)$ is the Jacobian matrix of $v(x)$ (see Appendix B for a definition of $\mathbf{J}(x)$).

It is easy to verify that if a game $\mathscr{G}$ is monotone, then it is also concave; however, the converse does not hold. We now turn our attention to the strict versions of these definitions.

**Definition 2.4** (Strictly Concave/Monotone Games). Consider a polynomial game $\mathscr{G}$ over a basic semialgebraic set $\mathcal{X}$. Then, $\mathscr{G}$ is strictly concave over $\mathcal{X}$ if

$$\mathbf{H}_{u_i}(x) \prec 0, \qquad \forall x \in \mathcal{X}, \quad i \in [\![n]\!]. \tag{12}$$

Furthermore, $\mathscr{G}$ is strictly monotone over $\mathcal{X}$ if $\mathbf{SJ}$ is negative definite on $\mathcal{X}$, i.e.,

$$\mathbf{SJ}(x) \prec 0, \qquad \forall x \in \mathcal{X}. \tag{13}$$

Finally, Rosen [49] also studied the class of diagonally strictly concave games, defined as those for which equality in (10) holds if and only if $x = x'$.

Several important connections and inclusions between the aforementioned game classes we study are summarized in Figure 1. The proofs for these inclusions follow directly from the definitions of the games and standard results from [48]. Of particular interest to us is the fact that, if $\mathscr{G}$ is strictly monotone (i.e., it satisfies (13)), then $\mathscr{G}$ is both diagonally strictly concave and strictly concave. Moreover, [49] also proved that diagonally strictly concave games admit a *unique* Nash equilibrium.

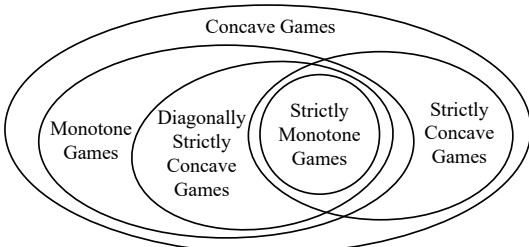

Figure 1: Connections and inclusions among the game classes we study.

# 3 Certifying Concavity and Monotonicity in Polynomial Games

As discussed in the introduction, concave and monotone games are highly expressive and have strong theoretical properties, including the existence of Nash equilibria, uniqueness under strictness conditions, and convergence of distributed dynamics to equilibrium. Given these favorable features, a natural question arises: can concavity/monotonicity be efficiently certified? In this section, we investigate this question in the setting of polynomial games with semialgebraic strategy sets.

To investigate hardness of deciding concavity/monotonicity, we leverage recent breakthroughs in polynomial optimization, particularly recent works on the complexity of certifying convexity of polynomials. Specifically, it has been shown in [4] that deciding whether a quartic (multivariate) polynomial is globally convex is NP-hard. Subsequently, [2] demonstrated that determining whether a cubic polynomial is convex over a box is also NP-hard. Building on these results, the starting point of this work is the observation that verifying whether a polynomial game belongs to the class of concave or monotone games is also NP-hard. This result is given below, and proven in Appendix C.1.

**Theorem 3.1.** *Let $\mathscr{G}([\![n]\!], \mathcal{X}, u)$ be a polynomial game over a compact convex basic semialgebraic set. If for some player $i$, $u_i$ is a polynomial of degree at least 3 with respect to $x_i \in \mathcal{X}_i$, verifying whether $\mathscr{G}$ is concave or monotone is strongly NP-hard.*

Motivated by the hardness result, it is crucial to identify tractable sufficient conditions for concavity and monotonicity, which gives rise to non-trivial subclasses of concave and monotone games. This can be achieved by using the technique of sum-of-squares optimization, together with the positive semidefiniteness of the Hessian or the symmetrized Jacobian matrix of the game. Throughout the remainder of the paper, for brevity *we focus only on the class of monotone games*. Analogous results hold for concave games with minor modifications, and we describe them in Section 5.

### 3.1 Sum-of-Squares Certificates for Concavity & Monotonicity

We introduce a hierarchy of increasingly strong sufficient conditions for certifying concavity and monotonicity, based on SOS certificates for the associated quadratic forms defined by the Hessian and the symmetrized Jacobian matrices of $\mathscr{G}$. The starting point for this observation is that, for any fixed $x \in \mathcal{X}$, and considering the symmetrized Jacobian, we have

$$\mathbf{SJ}(x) \preceq 0 \qquad \text{if and only if} \qquad \lambda_{\max}\big(\mathbf{SJ}(x)\big) \leq 0. \tag{14}$$

Consequently, using the *Rayleigh–Ritz theorem*, it follows that $\mathscr{G}$ is monotone if and only if

$$\max_{x \in \mathcal{X}} \lambda_{\max}\big(\mathbf{SJ}(x)\big) = \max_{\substack{x \in \mathcal{X} \\ y \in \mathcal{B}}} y^\mathsf{T} \mathbf{SJ}(x) y \leq 0. \tag{15}$$

where $\mathcal{B} \overset{\text{def}}{=} \big\{ y \in \mathbb{R}^m \mid y^\mathsf{T} y = 1 \big\}$. The crucial observation here is that the function $(x, y) \mapsto y^\mathsf{T} \mathbf{SJ}(x) y$ is a polynomial in $x, y$, since the Jacobian matrix $\mathbf{SJ}(x)$ is polynomial in $x$. Moreover, $\mathcal{X}$ and $\mathcal{B}$ are compact basic semialgebraic sets. Therefore, $\max_{x \in \mathcal{X}} \lambda_{\max}\big(\mathbf{SJ}(x)\big)$ can be written as the solution to the following polynomial maximization problem:

$$\max_{x \in \mathcal{X}} \lambda_{\max}\big(\mathbf{SJ}(x)\big) = \underset{x, y}{\text{maximize}} \qquad y^\mathsf{T} \mathbf{SJ}(x) y \tag{16}$$
$$\text{subject to} \qquad x \in \mathcal{X}, y \in \mathcal{B}.$$

Finally, although polynomial optimization is in general NP-hard, the solution to a polynomial optimization problem, i.e., $\max_{x \in \mathcal{X}} \lambda_{\max}\big(\mathbf{SJ}(x)\big)$ can be approximated via the SOS framework. This is formally stated in the main theorem of this section, the proof of which is given in Appendix C.2:

**Theorem 3.2.** *Let $\mathscr{G}(\llbracket n \rrbracket, \mathcal{X}, u)$ be a polynomial game over a compact, convex basic semialgebraic set $\mathcal{X}$. Assume the quadratic module $Q(\mathcal{X})$ is Archimedean. For any $\ell \in \mathbb{N}$ consider the hierarchy of SOS optimization problems:*

$$\mathrm{SOS}_\ell(\mathscr{G}) \overset{\text{def}}{=} \underset{\lambda \in \mathbb{R}}{\text{minimize}} \quad \lambda \tag{17}$$
$$\text{subject to} \quad \lambda - y^\mathsf{T} \mathbf{SJ}(x) y \in Q_\ell(\mathcal{X} \times \mathcal{B}),$$

*where $Q_\ell(\mathcal{X} \times \mathcal{B})$ denotes the restriction of $Q(\mathcal{X} \times \mathcal{B})$ to polynomials of degree at most $\ell$. Then, the following statements are true:*

1) *For all $\ell$, we have that $\mathrm{SOS}_\ell(\mathscr{G}) \geq \max_{x \in \mathcal{X}} \lambda_{max}\big(\mathbf{SJ}(x)\big)$.*

2) *The sequence $\big(\mathrm{SOS}_\ell(\mathscr{G})\big)_{\ell \geq 0}$ is nonincreasing.*

3) *$\lim_{\ell \to \infty} \mathrm{SOS}_\ell(\mathscr{G}) = \max_{x \in \mathcal{X}} \lambda_{max}\big(\mathbf{SJ}(x)\big)$.*

4) *$\mathscr{G}$ is strictly monotone if, and only if, there exists some finite level $\ell$ such that $\mathrm{SOS}_\ell(\mathscr{G}) < 0$.*

5) *For any level $\ell$, the program in (17) can be formulated as an semidefinite program (SDP) and solved in polynomial time.*

Theorem 3.2 shows how a sequence of SDPs, which can be solved efficiently (Statement 5), can be used to approximate $\max_{x \in \mathcal{X}} \lambda_{\max}\big(\mathbf{SJ}(x)\big)$, and therefore certify whether $\mathscr{G}$ is monotone. In particular, Statements 1 to 3 guarantee that $\mathrm{SOS}_\ell(\mathbf{SJ})$, for $\ell \geq 0$, gives progressively tighter upper bounds for $\max_{x \in \mathcal{X}} \lambda_{\max}\big(\mathbf{SJ}(x)\big)$. If for any finite $\ell$ we obtain $\mathrm{SOS}_\ell(\mathbf{SJ}) \leq 0$, it follows that $\max_{x \in \mathcal{X}} \lambda_{\max}\big(\mathbf{SJ}(x)\big) \leq 0$, and therefore $\mathscr{G}$ is monotone. Additionally, if at some $\ell$ we get $\mathrm{SOS}_\ell(\mathbf{SJ}) < 0$, it follows that $\max_{x \in \mathcal{X}} \lambda_{\max}\big(\mathbf{SJ}(x)\big) < 0$, and therefore $\mathscr{G}$ is strictly monotone.

Importantly, Statement 3 guarantees that whenever $\mathscr{G}$ is monotone, even if no finite $\ell$ exists such that $\mathrm{SOS}_\ell(\mathbf{SJ}) \leq 0$, the sequence $\big(\mathrm{SOS}_\ell(\mathbf{SJ})\big)_{\ell \geq 0}$ nonetheless converges (asymptotically) to a non-positive value. Moreover, whenever $\mathscr{G}$ is not only monotone but also strictly monotone, by Statement 4 we are guaranteed the existence of a finite $\ell$. In fact, it turns out that generic monotone polynomial games over compact, convex semialgebraic sets are *almost always* strictly monotone. In particular, in the following theorem we show that for all $\mathscr{G}$ of degree at least 2, the set of polynomial monotone games that are not strictly monotone form a set with zero Lebesgue measure.

**Theorem 3.3.** *For almost all monotone games, monotonicity can be certified at a finite level $\ell$ of the SOS hierarchy* (17)*, i.e., $\mathrm{SOS}_\ell(\mathscr{G}) \leq 0$. Concretely, for all $d \geq 2$, the set of monotone polynomial games of degree $d$ over a compact basic semialgebraic set $\mathcal{X}$ that are not strictly monotone has zero Lebesgue measure.*

The proof of this result is given in Appendix C.3. At this point, we have shown that the monotonicity of *almost all* polynomial monotone games $\mathscr{G}$ over a compact, convex semialgebraic set can be certified by a solution $\mathrm{SOS}_{\ell_\mathscr{G}}(\mathbf{SJ})$ at some finite level $\ell_\mathscr{G}$ of the SOS hierarchy in (17). However, for an arbitrary game $\mathscr{G}$, the required level $\ell_\mathscr{G}$ may be *large*. Thus, in practice, certifying the monotonicity of $\mathscr{G}$ via the SOS hierarchy in (17) may be computationally infeasible. To reflect this limitation, in the following section, we introduce and study a subclass of monotone games called $\ell$-SOS-monotone games, for which monotonicity can be certified in polynomial time via semidefinite programming.

## 4   SOS-Concave & SOS-Monotone Games

Motivated by the convergence guarantees of the SOS hierarchy established in Theorem 3.2, in this section, we define and analyze a subclass of polynomial monotone games over a compact, convex basic semialgebraic set for which monotonicity can be certified at some fixed level $\ell$ of the SOS hierarchy. These are games whose monotonicity can be verified in polynomial time with respect to the level $\ell$. We refer to such games as $\ell$-SOS-monotone.

**Definition 4.1** ($\ell$-SOS-Monotone Game)**.**  Consider a polynomial game $\mathscr{G} \in \mathcal{G}_{(n,d)}$ over a compact, convex basic semialgebraic set $\mathcal{X}$. For all $\ell \geq 0$, we say that $\mathscr{G}$ is $\ell$-SOS-monotone if

$$-y^{\mathsf{T}}\mathbf{SJ}(x)y \in Q_\ell(\mathcal{X} \times \mathcal{B}). \tag{18}$$

We denote the set of $\ell$-SOS-monotone games by $\mathcal{G}_{\mathrm{sosm}(n,d,\ell)}$. Furthermore, we say that $\mathscr{G}$ is SOS-monotone if there exists $\ell \in \mathbb{N}$ such that $\mathscr{G}$ is $\ell$-SOS-monotone.

The following theorem is an immediate consequence of Statement 4 in Theorem 3.2 and the measure-theoretic result in Theorem 3.3:

**Theorem 4.2.** *For all $d \geq 2$, the set of monotone polynomial games of degree $d$ over a compact, convex basic semialgebraic set $\mathcal{X}$ that are not SOS-monotone has zero Lebesgue measure.*

Next, we show that for every $\ell \geq 0$, the set of $\ell$-SOS-monotone games is a *global approximator* to the set of monotone games, i.e., SOS-monotone games are dense in monotone games. In particular, given some polynomial game $\mathscr{G}^*$ over a convex, compact basic semialgebraic set, we can compute the closest $\ell$-SOS-monotone game $\mathscr{G}$ in polynomial time. Moreover, since SOS-monotone games are dense in monotone games, as $\ell \to \infty$, the projections $\mathscr{G}$ of $\mathscr{G}^*$ in the set of $\ell$-SOS-monotone games converge to the closest monotone game to $\mathscr{G}^*$; not just the closest SOS-monotone game. The proof of the following theorem can be found in Appendix C.4.

**Theorem 4.3.** *For all $d \geq 2$, the set of SOS-monotone games of degree $d$ over a compact basic semialgebraic set $\mathcal{X}$ is dense in the set of monotone games of degree $d$ over $\mathcal{X}$. Furthermore, given any polynomial game $\mathscr{G}^* \in \mathcal{G}_{(n,d)}$ over $\mathcal{X}$, and any fixed $\ell \geq 0$, we can compute the closest $\ell$-SOS-monotone game to $\mathscr{G}^*$ by the program*

$$\begin{aligned}
\underset{\mathscr{G} \in \mathcal{G}_{(n,d)}}{\text{minimize}} \quad & \|\mathscr{G} - \mathscr{G}^*\| \\
\text{subject to} \quad & \mathscr{G} \in \mathcal{G}_{\mathrm{sosm}(n,d,\ell)},
\end{aligned} \tag{19}$$

*which can be formulated as an SDP.*

In Theorem 4.3 and throughout our experiments, distance between games is measured via the norm $\|\cdot\|$ in Eq. (3). Beyond the aforementioned norm, the optimization framework in Theorem 4.3 extends naturally to any function $\|\cdot\|$ that measures deviations on $\mathcal{G}_{(n,d)}$, whose epigraph is semidefinite representable and for which $\|\mathscr{G}_k - \mathscr{G}\| \to 0$ as $k \to \infty$, for all monotone games $\mathscr{G}$ and some sequence of SOS-monotone games.

For example, we give another example of a valid deviation operator. Let $\mathscr{G}^{\mathrm{quad}}(\llbracket n \rrbracket, \mathcal{X}, u^{\mathrm{quad}})$ be the SOS-monotone game with the payoff functions $u_i^{\mathrm{quad}}(x) = -\|x_i\|_2^2$, for all $i \in \llbracket n \rrbracket$. The *gauge* is given by

$$\gamma_\ell\big(\mathscr{G}(\llbracket n \rrbracket, \mathcal{X}, u)\big) \stackrel{\mathrm{def}}{=} \min\big\{\varepsilon \geq 0 \mid \mathscr{G} + \varepsilon \cdot \mathscr{G}^{\mathrm{quad}} \in \mathcal{G}_{\mathrm{sosm}(n,d,\ell)}\big\}, \tag{20}$$

where for all $\varepsilon \geq 0$, $\mathscr{G} + \varepsilon \cdot \mathscr{G}^{\mathrm{quad}}$ denotes the polynomial game $\mathscr{G}'(\llbracket n \rrbracket, \mathcal{X}, u')$ with the payoff functions $u_i'(x) = u_i(x) + \varepsilon \cdot u_i^{\mathrm{quad}}(x)$. Furthermore, the corresponding SDP is given by the $\ell$-th level of the SOS hierarchy in Theorem 3.2.

## 5 Modifications for the Certification of Concavity

The results in Sections 3.1 and 4 can be equivalently stated in relation to concave polynomial games over a compact, convex basic semialgebraic set, subject to minor modifications. By definition, a polynomial game $\mathscr{G}$ over a compact, convex basic semialgebraic set $\mathcal{X}$ is concave if and only if the Hessian matrices $\mathbf{H}_{u_i}(x)$ are negative semidefinite, for all $x \in \mathcal{X}$ and $i \in \llbracket n \rrbracket$. Furthermore, $\mathscr{G}$ is strictly concave if and only if $\mathbf{H}_{u_i}(x)$ are all negative definite. For each $i$, consider the SOS hierarchy $\big(\mathrm{SOS}_{i,\ell}(\mathscr{G})\big)_{\ell \geq 0}$ given in Eq. (17), where we substitute $\mathbf{SJ}(x)$ with $\mathbf{H}_{u_i}(x)$. Then, the SOS-based hierarchy

$$\big(\max_{i \in \llbracket n \rrbracket} \mathrm{SOS}_{i,\ell}(\mathscr{G})\big)_{\ell \geq 0} \tag{21}$$

provides analogous guarantees as in the case of monotone games. In particular, Theorem 3.2 and Theorem 4.3, as well as Definition 4.1 can be written analogously with respect to the SOS-based hierarchy in Eq. (21). Meanwhile, Theorems 3.3 and 4.2 can be written for concave games directly without further modifications. For completeness, we provide the definition of $\ell$-SOS-Concave games here:

**Definition 5.1** ($\ell$-SOS-Concave Game). Consider a polynomial game $\mathscr{G} \in \mathcal{G}_{(n,d)}$ over a compact, convex basic semialgebraic set $\mathcal{X}$. For all $\ell \geq 0$, we say that $\mathscr{G}$ is $\ell$-SOS-concave if

$$-y^\mathsf{T} \mathbf{H}_{u_i}(x) y \in Q_\ell(\mathcal{X} \times \mathcal{B}), \quad \forall i \in \llbracket n \rrbracket. \tag{22}$$

## 6 Application: Extensive-Form Games with Imperfect Recall

As described concisely in [20], the class of *concave* games has many modern applications. Similarly, *monotone* games have been studied extensively due to their desirable equilibrium properties (see e.g. [38, 19, 9] and references therein). In this section, we highlight extensive-form games (EFGs) with imperfect recall, leveraging the fact that they can be viewed as polynomial games over compact, convex basic semialgebraic sets. We also utilize our theoretical results and proposed game classes to study canonical examples of these games. We will defer further discussion on applications to *economic markets* to Appendix F.

The study of extensive-form or sequential games is arguably as classical as that of normal-form games. The reader is referred to [44, Sections II and III] for a review of standard concepts. Moreover, for the sake of notational brevity and readability, we defer formal definitions of EFGs and related concepts to Appendix D. One of the most important results in extensive-form games is Kuhn's theorem [33], which establishes a connection between mixed strategies and behavioral strategies in EFGs with perfect recall (wherein players effectively never forget the history of information sets visited and actions played). Relaxing the perfect recall assumption results in games where players can forget prior information, which introduces additional computational challenges.

The canonical example of an imperfect recall game is that of the absent-minded taxi driver (Figure 3), introduced in [46]. Furthermore, [46] showed that the expected utility of any player in an EFG with imperfect recall can be written as a polynomial, where each variable is associated with an information set (i.e., a collection of decision nodes which a player cannot distinguish between). In

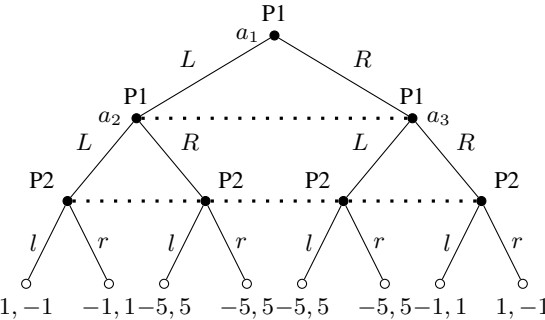
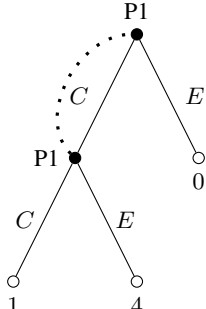

Figure 2: A Game with No Nash Equilibria     Figure 3: The Absent-minded Taxi Driver

particular, these utilities define an $n$-variable polynomial game $\mathscr{G}(\llbracket n \rrbracket, \mathcal{X}, u)$ over the simplex. For clarity, we derive the corresponding polynomial utility function of the game in Figure 3. Since the player has imperfect recall and cannot remember if they are in the first or second decision node, they will select a distribution over $\{C, E\}$ to be applied to both decision nodes. If the player selects $C$ with probability $x_1$ and $E$ with probability $x_2$, then their expected payoff is given by $x_1^2 + 4x_1 x_2$.

In general, though, a Nash equilibrium might not exist in EFGs with imperfect recall. For instance, the game in Figure 2 was introduced by [64] and does *not* have a Nash equilibrium. Several recent works have further established hardness results for deciding the existence of or computing NE in EFGs with imperfect recall [31, 59, 60, 23]. Theorem C.1 additionally guarantees the hardness of *verifying* concavity/monotonicity of EFGs with imperfect recall over simplex action sets.

## 6.1 Experimental Methodology & Results

Our results in Sections 3.1 and 4 motivate two lines of investigation: verifying monotonicity/concavity and computing the closest SOS-monotone/concave game.

**Examples 1 & 2.** First, we use the SDP hierarchy in Eq. (17) to certify SOS-monotonicity of the absent-minded taxi driver game in Figure 3. Next, we use the program in Eq. (19) to find an SOS-monotone game $\mathscr{G}$ which is closest to the zero-sum game in Figure 2 (which does not have a NE in behavioral strategies), in the sense of the norm defined in Eq. (3). By further enforcing that the new game has to be zero-sum and the monomial basis of the game is maintained, we are able to obtain the closest SOS-monotone game $\mathscr{G}'$ given by:

$$u_1'(x_1, x_2, y) = -8x_1 y - 8x_2 y - 16x_1 - 16x_2 - 12y - 9,$$

and $u_2' = -u_1'$. The distance between the two games is $\|\mathscr{G} - \mathscr{G}'\| = 10$, and since the modified game is SOS-monotone, it has a NE in behavioral strategies.

The above examples are applied to canonical EFGs with imperfect recall—going forward, we utilize our framework to study larger EFGs and aim to study the scalability of our approach. For brevity, full experimental details are deferred to Appendix E.

**Example 3: A degree-4 strictly monotone general-sum game.** [4, Theorem 2.3] introduces a method to construct (strictly) convex polynomials of degree 4. Using this method, we construct a two-player game with degree-4 polynomial utility functions that is strictly concave. P1 and P2 choose their actions $(x_1, x_2)$ and $(y_1, y_2)$ from a two-dimensional simplex respectively. By running our hierarchy of SOS optimization problems in Eq. (17) for monotonicity, we obtain an objective value $-1$ at level 4, thus certifying that the game is strictly monotone and also SOS-monotone.

**Example 4: A degree-5 zero-sum game.** We create a two-player zero-sum EFG with imperfect recall as shown in Figure E.4, where the payoffs on each leaf are for P1. In this example, P1 makes four moves before P2 makes a move, and P1 is absent-minded. By letting $x$ denote the probability that P1 chooses $L$ and $y$ denote the probability that P2 chooses $l$, we obtain the payoffs for P1 and P2 as follows:

$$u_1(x, y) = -16x^4 y + 25x^4 + 74x^3 y - 59x^3 - 89x^2 y + 49x^2 + 45xy - 19x - 8y + 3,$$

and $u_2 = -u_1$. We run our program in Eq. (19) to find the closest SOS-monotone game. Two additional constraints are imposed to retain the properties of the original EFG: The modified game has to be zero-sum, and the information structure of the original EFG has to be preserved. To preserve the information structure of the game, we select the monomial basis for the new payoff functions to be precisely the monomial basis that can appear in the original game. The following modified payoff functions are found:

$$u_1'(x, y) = -5.6x^4y - 6x^4 + 32.8x^3y - 22.9x^3 - 75.1x^2y - 4xy - 68x - 57y - 46,$$

and $u_2' = -u_1'$, with $\|\mathscr{G} - \mathscr{G}'\| = 49$.

**Example 5: A degree-8 general-sum game.** We construct a two-player EFG with imperfect recall where P1 makes six moves before P2 makes two moves. There is one information set for P1 and one information set for P2. P1 has three actions to choose from with probability $x_1$, $x_2$, and $1 - x_1 - x_2$, respectively. P2 also has three actions to choose from with probability $y_1$, $y_2$, and $1 - y_1 - y_2$, respectively. Hence, the game tree has nine layers, including the root and the leaves, and the payoff functions are degree-8 polynomials with monomial basis

$$[x_1^6, x_1^5x_2, x_1^4x_2^2, x_1^3x_2^3, x_1^2x_2^4, x_1x_2^5, x_2^6, x_1^5, x_1^4x_2, x_1^3x_2^2, x_1^2x_2^3, x_1x_2^4, x_2^5, x_1^4, x_1^3x_2, x_1^2x_2^2,$$
$$x_1x_2^3, x_2^4, x_1^3, x_1^2x_2, x_1x_2^2, x_2^3, x_1^2, x_1x_2, x_2^2, x_1, x_2, 1] \otimes [y_1^2, y_1y_2, y_2^2, y_1, y_2, 1],$$

where $\otimes$ is the tensor product. The size of the monomial basis is 168. We do not restrict the EFG to be zero-sum, but instead randomly generate the payoff functions for P1 and P2 by independently sampling the coefficient of each monomial in the basis from a uniform distribution on $[-1, 1]$.

We run our program in Eq. (19) to find the closest SOS-monotone game with the additional constraint that the information structure of the original EFG has to be preserved, i.e. the new payoff functions have to be polynomials with the same monomial basis. As in Example 3, we defer the full payoff functions of the game to Appendix E.

**On Scalability.** A natural limitation of our framework is scalability—while SDPs can be solved with arbitrary accuracy in polynomial time using interior point methods, they are among the most expensive convex relaxations to solve. In practice, "SOS problems involving degree-4 or 6 polynomials are currently limited, roughly speaking, to a handful or a dozen variables" [3]. We compare the compute times of our proposed hierarchies when applied to the larger-scale examples above. Indeed, while the SOS hierarchies in Examples 3 and 4 can be solved in $\approx 0.052$ and $\approx 0.009$ seconds respectively, the much larger program for Example 5 took $\approx 37.53$ seconds to solve using a standard, off-the-shelf solver. This further motivates future work on scaling our approach using existing methods in the literature [3, 37, 66, 39, 27]. Our code[1] is implemented using the SumOfSquares package for Julia [36, 63] and run on a MacBook Air with 16 GB RAM.

# 7 Discussion

In this paper, we have shown that verifying concavity and monotonicity in polynomial games is in general NP-hard. For polynomial games over compact, convex basic semialgebraic sets, we utilize SOS techniques to construct SDP hierarchies that can certify concavity and monotonicity. Moreover, we show that almost all concave/monotone games are strict, and thus can be certified at a finite level of the respective hierarchy. Finally, we introduced $\ell$-SOS-concave and $\ell$-SOS-monotone games, which are certified at some fixed level $\ell$ of the respective SOS hierarchy. This leads to an application for EFGs of imperfect recall, where we are able to find the closest (in terms of an appropriate norm) SOS-concave/monotone game to a canonical EFG which has no Nash equilibria. In addition, in light of the experiments in Section 6.1, our work motivates the design of application-specific programs which can find close concave/monotone games while also maintaining structural properties of the original game.

**Broader Impact.** While our results are primarily theoretical, we acknowledge that there could be potential societal consequences of our work, none of which we feel must be specifically highlighted.

---

[1]Code used to generate the experiments in Section 6 can be found in our github repo.

**Acknowledgements** This work is supported by the MOE Tier 2 Grant (MOE-T2EP20223-0018), Ministry of Education Singapore (SRG ESD 2024 174), the CQT++ Core Research Funding Grant (SUTD) (RS-NRCQT-00002), the National Research Foundation Singapore and DSO National Laboratories under the AI Singapore Programme (Award Number: AISG2-RP-2020-016), and partially by Project MIS 5154714 of the National Recovery and Resilience Plan, Greece 2.0, funded by the European Union under the NextGenerationEU Program. The authors also thank anonymous reviewers for their insightful feedback during the review process.

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

## A   Additional Related Work

**Polynomial Games and Semidefinite Programming.**   Initially introduced and studied by [14], polynomial games were viewed as a bridge between finite and continuous games. Although [14] characterized and proved the existence of equilibria in these games, providing computational guarantees for general polynomial games has proven to be a challenging task. [45, 34] used semidefinite programming methods to find the value of two-player zero-sum polynomial games, and similar techniques apply to separable games (where utilities take a sum-of-products form) [55]. Recently, [43] also used semidefinite programming techniques to solve for Nash equilibria in $n$-player polynomial games, or otherwise detect the nonexistence of equilibria. Beyond polynomial games, oracle-based methods have been used to approximately solve continuous games [1, 32].

## B   Additional Preliminaries

Given a game $\mathscr{G}$ with utility functions $u_i(x)$, the standard Jacobian is defined as follows:

$$\mathbf{J}(x) \overset{\text{def}}{=} \begin{pmatrix} \mathbf{H}_{u_1}(x) & \nabla_{x_2}^{\mathsf{T}}(\nabla_{x_1} u_1)(x) & \dots & \nabla_{x_n}^{\mathsf{T}}(\nabla_{x_1} u_1)(x) \\ \nabla_{x_1}^{\mathsf{T}}(\nabla_{x_2} u_2)(x) & \mathbf{H}_{u_2}(x) & \dots & \nabla_{x_n}^{\mathsf{T}}(\nabla_{x_2} u_2)(x) \\ \vdots & \vdots & \ddots & \vdots \\ \nabla_{x_1}^{\mathsf{T}}(\nabla_{x_n} u_n)(x) & \nabla_{x_2}^{\mathsf{T}}(\nabla_{x_n} u_n)(x) & \dots & \mathbf{H}_{u_n}(x) \end{pmatrix}. \tag{A23}$$

## C   Omitted Proofs from Main Text

### C.1   Proof of Theorem 3.1

As mentioned earlier, several works have studied the hardness of verifying convexity in multivariate polynomials [4, 2]. We state the main theorem for hardness of verifying convexity over a *box* here for completeness:

**Theorem C.1** ([2, Theorem 2.3]). *Deciding whether a polynomial of degree at least $3$ is convex over a box $\mathcal{D} \overset{\text{def}}{=} \{x \in \mathbb{R}^n \mid \alpha_i \leq x_i \leq \beta_i \quad i \in [\![n]\!]\}$, for $\alpha, \beta \in \mathbb{R}^n$, is strongly NP-hard.*

Using this result, we are able to prove NP-hardness of verifying concavity and monotonicity in polynomial games.

**Theorem 3.1.** *Let $\mathscr{G}([\![n]\!], \mathcal{X}, u)$ be a polynomial game over a compact convex basic semialgebraic set. If for some player $i$, $u_i$ is a polynomial of degree at least $3$ with respect to $x_i \in \mathcal{X}_i$, verifying whether $\mathscr{G}$ is concave or monotone is strongly NP-hard.*

*Proof.* To prove the statement for concave games, let $p \colon \mathbb{R}^m \to \mathbb{R}$ be a polynomial of degree 3 or higher, and let $\mathcal{D} \overset{\text{def}}{=} \{x \in \mathbb{R}^m \mid \alpha_j \leq x_j \leq \beta_j \quad j \in [\![m]\!]\}$, for $\alpha, \beta \in \mathbb{R}^n$ and $\alpha_j \leq \beta_j$, be a box over $\mathbb{R}^m$. Consider a two-player polynomial game $\mathscr{G}([\![2]\!], \mathcal{D} \times \mathcal{D}, u)$, where $u_1(x) = p(x_1)$, and $u_2(x) = 0$ for all $x \in \mathcal{D} \times \mathcal{D}$. Then, since $u_2$ is concave over $\mathcal{D}$, the game $\mathscr{G}$ is concave if and only if $p$ is concave over $\mathcal{D}$. Thus, from Theorem C.1 it follows that verifying whether $\mathscr{G}$ is concave is strongly NP-hard.

To prove the statement for monotone games, let $p \colon \mathbb{R}^m \to \mathbb{R}$ be a polynomial of degree 3 or higher, and let $\mathcal{D} \overset{\text{def}}{=} \{x \in \mathbb{R}^m \mid \alpha_j \leq x_j \leq \beta_j \quad j \in [\![m]\!]\}$, for $\alpha, \beta \in \mathbb{R}^n$ and $\alpha_j \leq \beta_j$, be a box over $\mathbb{R}^m$. Note that the description of $\mathcal{D}$ satisfies Eq. (1), and hence $\mathcal{D}$ is a convex and compact semialgebraic set. Define $\mathcal{X}_1 \overset{\text{def}}{=} [\alpha_1, \beta_1] \times \cdots \times [\alpha_{m-1}, \beta_{m-1}]$, and $\mathcal{X}_2 \overset{\text{def}}{=} [\alpha_m, \beta_m]$. Clearly, $\mathcal{X}_1 \times \mathcal{X}_2 = \mathcal{D}$. Consider a two-player polynomial game $\mathscr{G}([2], \mathcal{X}_1 \times \mathcal{X}_2, u)$, where $u_1(x) = u_2(x) = p(x)$ for all $x \in \mathcal{X}$. Then, the game $\mathscr{G}$ is monotone if and only if the operator $\left(-\nabla_{x_1} u_1(x), -\nabla_{x_2} u_2(x)\right) = -\nabla p(x)$ is monotone over $\mathcal{X}_1 \times \mathcal{X}_2$, or equivalently $p$ is concave over $\mathcal{D}$. Thus, from Theorem C.1 it follows that verifying whether $\mathscr{G}$ is a monotone game is strongly NP-hard. $\square$

As a direct consequence of the hardness of verifying the concavity of degree-4 polynomials over the simplex [26, 4], we can obtain the following NP-hardness result for verifying concavity/monotonicity in polynomial games with simplex action sets.

**Theorem C.2.** *Let $\mathcal{G}(\llbracket n \rrbracket, \mathcal{X}, u)$ be a polynomial game where $\mathcal{X}_i \in \Delta^{m_i}$. If for some player $i$, $u_i$ is a polynomial of degree at least 4 with respect to $x_i \in \mathcal{X}_i$, then verifying whether $\mathcal{G}$ is concave/monotone is strongly NP-hard.*

*Proof.* We provide a similar construction to the proof of Theorem 3.1. Here we show hardness for verifying concavity – the proof for hardness of verifying monotonicity is similar. Let $p\colon \mathbb{R}^m \to \mathbb{R}$ be a polynomial of degree 4 or higher, and let $\Delta \stackrel{\text{def}}{=} \{x \in \mathbb{R}^m \mid \sum_{i \in \llbracket m \rrbracket} x_i = 1, x_i \geq 0$ for $i = 1, \ldots, m\}$ be the $m$-dimensional simplex. Consider a two-player polynomial game $\mathcal{G}(\llbracket 2 \rrbracket, \Delta \times \Delta, u)$, where $u_1(x) = p(x_1)$, and $u_2(x) = 0$ for all $x \in \Delta \times \Delta$. $u_2$ is concave over $\Delta$, so the game $\mathcal{G}$ is concave if and only if $p$ is concave over $\Delta$. Thus, from [4, Theorem 2.1] (and as indirectly argued in [26] utilizing [40, Theorem 1]), it follows that verifying whether $\mathcal{G}$ is concave is strongly NP-hard. $\qquad\square$

## C.2  Proof of Theorem 3.2

**Theorem 3.2.** *Let $\mathcal{G}(\llbracket n \rrbracket, \mathcal{X}, u)$ be a polynomial game over a compact, convex basic semialgebraic set $\mathcal{X}$. Assume the quadratic module $Q(\mathcal{X})$ is Archimedean. For any $\ell \in \mathbb{N}$ consider the hierarchy of SOS optimization problems:*

$$\text{SOS}_\ell(\mathcal{G}) \stackrel{\text{def}}{=} \begin{array}{cc} \underset{\lambda \in \mathbb{R}}{\text{minimize}} & \lambda \\ \text{subject to} & \lambda - y^\mathsf{T} \mathbf{SJ}(x)y \in Q_\ell(\mathcal{X} \times \mathcal{B}), \end{array} \tag{17}$$

*where $Q_\ell(\mathcal{X} \times \mathcal{B})$ denotes the restriction of $Q(\mathcal{X} \times \mathcal{B})$ to polynomials of degree at most $\ell$. Then, the following statements are true:*

1) *For all $\ell$, we have that $\text{SOS}_\ell(\mathcal{G}) \geq \max_{x \in \mathcal{X}} \lambda_{max}(\mathbf{SJ}(x))$.*

2) *The sequence $(\text{SOS}_\ell(\mathcal{G}))_{\ell \geq 0}$ is nonincreasing.*

3) *$\lim_{\ell \to \infty} \text{SOS}_\ell(\mathcal{G}) = \max_{x \in \mathcal{X}} \lambda_{max}(\mathbf{SJ}(x))$.*

4) *$\mathcal{G}$ is strictly monotone if, and only if, there exists some finite level $\ell$ such that $\text{SOS}_\ell(\mathcal{G}) < 0$.*

5) *For any level $\ell$, the program in (17) can be formulated as an SDP and solved in polynomial time.*

*Proof.* To prove Statement 1, we start by considering some arbitrary $\ell \geq 0$. Observe that, if the program in (17) is infeasible, then $\text{SOS}_\ell(\mathbf{SJ}) = \infty$, and therefore $\text{SOS}_\ell(\mathbf{SJ}) \geq \lambda_{\max}(\mathbf{SJ}(x))$ for all $x \in \mathcal{X}$. On the other hand, if the program in (17) is feasible, $\text{SOS}_\ell(\mathbf{SJ}) - y^\mathsf{T} \mathbf{SJ}(x)y \in Q_\ell(\mathcal{X} \times \mathcal{B})$, i.e., there exist $\sigma_1^*, \ldots, \sigma_{m_g}^* \in \Sigma[x]$ and $p_0^*, \ldots, p_{m_h}^* \in \mathbb{R}[x]$ such that

$$\text{SOS}_\ell(\mathbf{SJ}) - y^\mathsf{T}\mathbf{SJ}(x)y = \sigma_0^*(x,y) + \sum_{j=1}^{m_g} g_j(x)\sigma_j^*(x,y) + (1 - y^\mathsf{T}y)p_0^*(x,y) + \sum_{j=1}^{m_h} h_j(x)p^*(x,y), \forall x, y \in \mathbb{R}^m.$$

$$\tag{A24}$$

In particular, since $h_1(x) = \cdots = h_{m_h}(x) = 0$ for all $x \in \mathcal{X}$; and $1 - y^\mathsf{T}y = 0$ for all $y \in \mathcal{B}$, by the above we also have that

$$\text{SOS}_\ell(\mathbf{SJ}) - y^\mathsf{T}\mathbf{SJ}(x)y = \sigma_0^*(x,y) + \sum_{j=1}^{m_g} g_j(x)\sigma_j^*(x,y) \geq 0, \qquad \forall x \in \mathcal{X},\, y \in \mathcal{B}, \tag{A25}$$

where the last inequality follows because $g_j(x) \geq 0$ for all $x \in \mathcal{X}$, and $\sigma_0, \ldots, \sigma_{m_g} \in \Sigma[x,y]$. Furthermore, since $\mathbf{SJ}$ is symmetric, the maximum eigenvalue of $\mathbf{SJ}(x)$ is given by

$$\lambda_{\max}(\mathbf{SJ}(x)) \stackrel{\text{def}}{=} \max_{y \in \mathcal{B}} y^\mathsf{T}\mathbf{SJ}(x)y \stackrel{(A25)}{\leq} \text{SOS}_\ell(\mathbf{SJ}), \qquad \forall x \in \mathcal{X}. \tag{A26}$$

Alas, we have established that, $\text{SOS}_\ell(\mathbf{SJ}) \geq \lambda_{\max}(\mathbf{SJ}(x))$ for all $x \in \mathcal{X}$ and $\ell \geq 0$.

To prove Statement 2, observe that as $\ell$ increases, the feasible set of the minimization program in (17) is expanding, and therefore $(\text{SOS}_\ell(\mathbf{SJ}))_{\ell \geq 0}$ is nonincreasing.

To prove Statement 3, we, first, prove that $\lim_{\ell \to \infty} \mathrm{SOS}_\ell(\mathbf{SJ})$ exists, i.e., the sequence $\big(\mathrm{SOS}_\ell(\mathbf{SJ})\big)_{\ell \geq 0}$ converges. In particular, observe that, since the quadratic module $Q(\mathcal{X})$ is Archimedean, the set $\mathcal{X}$ is compact, and thus, the maximum $\max_{x \in \mathcal{X}} \lambda_{\max}\big(\mathbf{SJ}(x)\big)$ exists. Then, by Statement 1, it follows that

$$\max_{x \in \mathcal{X}} \lambda_{\max}\big(\mathbf{SJ}(x)\big) \leq \mathrm{SOS}_\ell(\mathbf{SJ}), \qquad \forall \ell \geq 0. \tag{A27}$$

Consequently, the sequence $\big(\mathrm{SOS}_\ell(\mathbf{SJ})\big)_{\ell \geq 0}$ is nonincreasing (Statement 2) and bounded from below by (Equation (A27)), and therefore it converges.

Now, recall that $\lim_{\ell \to \infty} \mathrm{SOS}_\ell(\mathbf{SJ}) = \max_{x \in \mathcal{X}} \lambda_{\max}\big(\mathbf{SJ}(x)\big)$ if for every $\epsilon > 0$, there exists $\ell_0 \geq 0$ such that $|\mathrm{SOS}_\ell(\mathbf{SJ}) - \max_{x \in \mathcal{X}} \lambda_{\max}\big(\mathbf{SJ}(x)\big)| \leq \epsilon$ for all $\ell \geq \ell_0$. We are going to use *Putinar's Positivstellensatz* to show that for every $\epsilon > 0$ such an $\ell_0$ exists.

First, observe that $\mathcal{X} \times \mathcal{B}$ is a basic semialgebraic set. In particular, we have that

$$\mathcal{X} \times \mathcal{B} \equiv \left\{ (x,y) \in \mathbb{R}^m \times \mathbb{R}^m \;\middle|\; \begin{array}{l} g'_j(x,y) \stackrel{\text{def}}{=} g_j(x) \geq 0, \; j \in [m_g], \\ h'_0(x,y) \stackrel{\text{def}}{=} 1 - y^\mathsf{T} y = 0, \\ h'_j(x,y) \stackrel{\text{def}}{=} h_j = 0, \; j \in [m_h] \end{array} \right\}. \tag{A28}$$

Furthermore, it is not difficult to show that the quadratic module $Q(\mathcal{X} \times \mathcal{B})$ is Archimedean.

Indeed, since $Q(\mathcal{X})$ is Archimedean, there exists $N \in \mathbb{N}$ such that $N - \sum_{i=1}^m x_i^2 \in Q(\mathcal{X})$. Therefore, there exist $\sigma_0, \ldots, \sigma_{m_g} \in \Sigma[x]$, and $p_1, \ldots, p_{m_h} \in \mathbb{R}[x]$ such that

$$N - \sum_{i=1}^m x_i^2 = \sigma_0(x) + \sum_{j=1}^{m_g} g_j(x)\sigma_j(x) + \sum_{j=1}^{m_h} h_j(x)p_j(x), \qquad \forall x \in \mathbb{R}^m. \tag{A29}$$

Define the polynomial functions $\sigma'_0, \ldots, \sigma'_{m_g}, p'_0, \ldots, p'_{m_h} : \mathbb{R}^m \times \mathbb{R}^m \to \mathbb{R}$ given by

$$\begin{aligned} \sigma'_j(x,y) &= \sigma_j(x) & j &= 0, \ldots, m_g \\ p'_0(x,y) &= 1 & & \qquad \forall x, y \in \mathbb{R}^m, \\ p'_j(x,y) &= p_j(x) & j &= 1, \ldots, m_h \end{aligned} \tag{A30}$$

and observe that, since $\sigma_0, \ldots, \sigma_{m_g} \in \Sigma[x]$, it follows that $\sigma'_0, \ldots, \sigma'_{m_g} \in \Sigma[x,y]$. Moreover, observe that

$$N + 1 - \sum_{i=1}^m x_i^2 - \sum_i^m y_i^2 = \sigma'_0(x) + \sum_{j=1}^{m_g} g'_j(x)\sigma'_j(x) + \sum_{j=0}^{m_h} h'_j(x)p'_j(x), \qquad \forall x, y \in \mathbb{R}^m, \tag{A31}$$

and therefore, $N + 1 - \sum_{i=1}^m x_i^2 - \sum_i^m y_i^2 \in Q(\mathcal{X} \times \mathcal{B})$. Thus, we conclude that $Q(\mathcal{X} \times \mathcal{B})$ is Archimedean.

Next, observe that, by definition, $\max_{x \in \mathcal{X}} \lambda_{\max}\big(\mathbf{SJ}(x)\big) = \max_{\substack{x \in \mathcal{X} \\ y \in \mathcal{B}}} y^\mathsf{T} \mathbf{SJ}(x)y$, which also implies that

$$\max_{x \in \mathcal{X}} \lambda_{\max}\big(\mathbf{SJ}(x)\big) - y^\mathsf{T} \mathbf{SJ}(x)y \geq 0, \qquad \forall x \in \mathcal{X}, \, y \in \mathcal{B}. \tag{A32}$$

Therefore, the polynomial

$$q_\epsilon(x,y) = \max_{x \in \mathcal{X}} \lambda_{\max}\big(\mathbf{SJ}(x)\big) - y^\mathsf{T} \mathbf{SJ}(x)y + \epsilon \tag{A33}$$

is *positive* over $\mathcal{X} \times \mathcal{B}$ for all $\epsilon > 0$. Thus, by Putinar's Positivstellensatz, it follows that $q_\epsilon(x,y) \in Q(\mathcal{X} \times \mathcal{B})$, i.e., there exist $\sigma_0, \ldots, \sigma_{m_g} \in \Sigma[x,y]$, and $p_0, \ldots, p_{m_h} \in \mathbb{R}[x,y]$ such that

$$\begin{aligned} \Big( \max_{x \in \mathcal{X}} \lambda_{\max}\big(\mathbf{SJ}(x)\big) + \epsilon \Big) - y^\mathsf{T} \mathbf{SJ}(x)y &= q_\epsilon(x,y) \\ &= \sigma_0(x,y) + \sum_{j=1}^{m_g} g_j(x)\sigma_j(x,y) \qquad\qquad \forall x, y \in \mathbb{R}^m. \\ &\quad + (1 - y^\mathsf{T} y)p_0(x,y) + \sum_{j=1}^{m_h} h_j(x)p_j(x,y) \end{aligned}$$

$$\tag{A34}$$

Let $\ell_0 \geq 0$ be the smallest number such that $2\ell_0 \geq \max\{\deg(\sigma_0), \ldots, \deg(\sigma_{m_g}), \deg(p_0), \ldots, \deg(p_{m_h})\}$. Then, it follows that $\left(\max_{x \in \mathcal{X}} \lambda_{\max}(\mathbf{SJ}(x)) + \epsilon, \sigma, p\right)$ is a solution to the SOS program in (17), where $\ell = \ell_0$. Thus, by the optimality of $\mathrm{SOS}_{\ell_0}(\mathbf{SJ})$, and since the sequence $(\mathrm{SOS}_\ell(\mathbf{SJ}))_{\ell \geq 0}$ is nonincreasing (Statement 1), it follows that

$$\mathrm{SOS}_\ell(\mathbf{SJ}) \leq \mathrm{SOS}_{\ell_0}(\mathbf{SJ}) \leq \max_{x \in \mathcal{X}} \lambda_{\max}(\mathbf{SJ}(x)) + \epsilon, \qquad \forall \ell \geq \ell_0. \tag{A35}$$

Alas, by (A27), we conclude that

$$|\mathrm{SOS}_\ell(\mathbf{SJ}) - \max_{x \in \mathcal{X}} \lambda_{\max}(\mathbf{SJ}(x))| = \mathrm{SOS}_\ell(\mathbf{SJ}) - \max_{x \in \mathcal{X}} \lambda_{\max}(\mathbf{SJ}(x)) \leq \epsilon, \qquad \forall \ell \geq \ell_0. \tag{A36}$$

To prove Statement 4, recall that a polynomial game is strictly monotone if, and only if,

$$\max_{x \in \mathcal{X}} \lambda_{\max}(\mathbf{SJ}(x)) < 0. \tag{A37}$$

First, suppose that $\mathscr{G}$ is not strictly monotone. Then, by the above, we have that $\max_{x \in \mathcal{X}} \lambda_{\max}(\mathbf{SJ}(x)) \geq 0$. Thus, by Statement 1, it follows that

$$\mathrm{SOS}_\ell(\mathbf{SJ}) \geq \max_{x \in \mathcal{X}} \lambda_{\max}(\mathbf{SJ}(x)) \geq 0, \qquad \forall \ell \geq 0. \tag{A38}$$

Next, suppose instead that $\mathscr{G}$ is strictly monotone. Then, $\max_{x \in \mathcal{X}} \lambda_{\max}(\mathbf{SJ}(x)) < 0$, and therefore, it exists $\epsilon > 0$ such that $\max_{x \in \mathcal{X}} \lambda_{\max}(\mathbf{SJ}(x)) + \epsilon < 0$. Moreover, by Statement 3, we also have that

$$\lim_{\ell \to \infty} \mathrm{SOS}_\ell(\mathbf{SJ}) = \max_{x \in \mathcal{X}} \lambda_{\max}(\mathbf{SJ}(x)). \tag{A39}$$

Therefore, by definition, there exists $\ell_0 \geq 0$ such that $|\mathrm{SOS}_{\ell_0}(\mathbf{SJ}) - \max_{x \in \mathcal{X}} \lambda_{\max}(\mathbf{SJ}(x))| \leq \epsilon$, and thus

$$\mathrm{SOS}_{\ell_0}(\mathbf{SJ}) \leq \max_{x \in \mathcal{X}} \lambda_{\max}(\mathbf{SJ}(x)) + \epsilon < 0. \tag{A40}$$

Statement 5 follows from standard results in semidefinite programming. $\qquad \square$

## C.3   Proof of Theorem 3.3

**Theorem 3.3.** *For almost all monotone games, monotonicity can be certified at a finite level $\ell$ of the SOS hierarchy (17), i.e., $\mathrm{SOS}_\ell(\mathscr{G}) \leq 0$. Concretely, for all $d \geq 2$, the set of monotone polynomial games of degree $d$ over a compact basic semialgebraic set $\mathcal{X}$ that are not strictly monotone has zero Lebesgue measure.*

*Proof.* Let $\mathcal{G}_{\mathrm{m}(n,d)}$ and $\mathcal{G}_{\mathrm{sm}(n,d)}$ denote the sets of $n$-player, $d$-degree polynomial monotone and strictly monotone games, respectively. We are going to show that given a compact, convex basic semialgebraic set $\mathcal{X}$ of joint actions, the set $\mathcal{G}_{\mathrm{m}(n,d)} \setminus \mathcal{G}_{\mathrm{sm}(n,d)}$ has zero Lebesgue measure. In particular, define $\mu$ as the canonical $\dim(\mathcal{G}_{\mathrm{m}(n,d)})$-dimensional Lebesgue measure on $\mathrm{aff}(\mathcal{G}_{\mathrm{m}(n,d)})$, i.e., the affine hull of $\mathcal{G}_{\mathrm{m}(n,d)}$. First, we show that $\mathcal{G}_{\mathrm{m}(n,d)}$ is $\mu$-measurable and therefore the restriction of $\mu$ to $\mathcal{G}_{\mathrm{m}(n,d)}$ (denoted by $\mu \upharpoonright_{\mathcal{G}_{\mathrm{m}(n,d)}}$) is well-defined. Then, we show that $\mu \upharpoonright_{\mathcal{G}_{\mathrm{m}(n,d)}} (\mathcal{G}_{\mathrm{m}(n,d)} \setminus \mathcal{G}_{\mathrm{sm}(n,d)}) = 0$.

To begin with, observe that by definition:

$$\mathcal{G}_{\mathrm{m}(n,d)} \equiv \left\{ \mathscr{G} \in \mathcal{G}_{(n,d)} \,\big|\, \mathbf{SJ}_{\mathscr{G}}(x) \succeq 0, \, \forall x \in \mathcal{X} \right\}, \tag{A41}$$

where for each $\mathscr{G} \in \mathcal{G}_{\mathrm{m}(n,d)}$, $\mathbf{SJ}_{\mathscr{G}}$ is the symmetrized Jacobian matrix of the pseudo-gradient $v_{\mathscr{G}}$ of $\mathscr{G}$. Next, observe that the map $(\mathscr{G}, x) \mapsto \mathbf{SJ}_{\mathscr{G}}(x)$ is polynomial in $\mathscr{G}$ and $x$. Moreover, the determinant $\mathbf{A} \mapsto \det(\mathbf{A})$ is also polynomial in $\mathbf{A}$. Therefore, for all $\mathscr{G} \in \mathcal{G}_{\mathrm{m}(n,d)}$ and $x \in \mathcal{X}$, the principal minors $f_{\mathcal{I}} \colon (\mathscr{G}, x) \mapsto \det(\mathbf{SJ}_{\mathscr{G},\mathcal{I}}(x))$, $\mathcal{I} \in 2^{[\![m]\!]}$, of $\mathbf{SJ}_{\mathscr{G}}(x)$ are polynomial in $\mathscr{G}$ and $x$. Thus, by *Sylvester's Criterion*:

$$\mathcal{G}_{\mathrm{m}(n,d)} \equiv \left\{ \mathscr{G} \in \mathcal{G}_{(n,d)} \,\big|\, f_{\mathcal{I}}(\mathscr{G}, x) \geq 0, \, \forall \mathcal{I} \in 2^{[\![m]\!]}, \, x \in \mathcal{X} \right\}. \tag{A42}$$

It follows by the *Tarski-Seidenberg theorem* [58, 52] that $\mathcal{G}_{\mathrm{m}(n,d)}$ is a basic semialgebraic set, and therefore a Borel set. Thus, $\mathcal{G}_{\mathrm{m}(n,d)}$ is $\mu$-measurable.

Next, observe that $\mathcal{G}_{\mathrm{m}(n,d)}$ is convex. Indeed, for all $x \in \mathcal{X}$, define

$$\mathcal{S}_x \overset{\text{def}}{=} \{\mathscr{G} \in \mathcal{G}_{(n,d)} \mid \mathbf{SJ}_{\mathscr{G}}(x) \succeq 0\}. \tag{A43}$$

Observe that the map $J_x \colon \mathscr{G} \mapsto \mathbf{SJ}_{\mathscr{G}}(x)$ linear. Since $\mathcal{G}_{(n,d)}$ is a vector space, $\mathcal{S}_x \equiv J_x(\mathcal{G}_{(n,d)})$ is a vector subspace, and therefore convex. Thus,

$$\mathcal{G}_{\mathrm{m}(n,d)} \equiv \bigcap_{x \in \mathcal{X}} \mathcal{S}_x \tag{A44}$$

is the (uncountable) intersection of convex sets, and therefore it is convex. Note that since the empty set is convex, the statement in Eq. (A44) remains valid even if the intersection is empty.

Let us now consider the set

$$\mathcal{G}_{\mathrm{sm}(n,d)} \equiv \left\{\mathscr{G} \in \mathcal{G}_{(n,d)} \;\middle|\; \mathbf{J}_{\mathscr{G}}(x) \succ 0, \; \forall x \in \mathcal{X}\right\}. \tag{A45}$$

Observe that for all $d \geq 0$, $\mathcal{G}_{\mathrm{sm}(n,d)}$ is non-empty as the game $\mathscr{G}'$ with payoff functions $u_i \colon x \mapsto \frac{1}{2}\|x_i\|^2$, for all $i \in [\![n]\!]$, is strictly monotone. We show that $\mathcal{G}_{\mathrm{sm}(n,d)} \supseteq \mathrm{int}(\mathcal{G}_{\mathrm{m}(n,d)})$ with respect to the relative topology.

Let $\mathscr{G}_0 \in \mathrm{int}(\mathcal{G}_{\mathrm{m}(n,d)})$, and suppose $\mathscr{G}_0 \notin \mathcal{G}_{\mathrm{sm}(n,d)}$. Then, by definition, there exists $x_0 \in \mathcal{X}$ such that $\mathbf{SJ}_{\mathscr{G}_0}(x) \not\succ 0$, i.e., it exists a vector $u \in \mathbb{R}^m \setminus \{0\}$ such that $u^\mathsf{T}\mathbf{SJ}_{\mathscr{G}_0}(x_0)u = 0$. Define $L \colon \mathscr{G} \mapsto u^\mathsf{T}\mathbf{SJ}_{\mathscr{G}}(x_0)u$. Since $J_{x_0}$ is a linear map, it follows that $L$ a linear functional. In particular, we have that $L(\mathscr{G}_0) = 0$. Moreover, since by definition $\mathbf{SJ}_{\mathscr{G}}(x) \succeq 0$ for all $\mathscr{G} \in \mathcal{G}_{\mathrm{m}(n,d)}$ and $x \in \mathcal{X}$, we also have that $L(\mathcal{G}_{\mathrm{m}(n,d)}) \geq 0$. Finally, since $\mathcal{G}_{\mathrm{sm}(n,d)}$ is non-empty, by definition we have that $L(\mathcal{G}_{\mathrm{sm}(n,d)}) > 0$, and therefore $L$ is non-trivial, i.e., $L \not\equiv 0$. Thus, $L$ describes a non-trivial supporting hyperplane to $\mathcal{G}_{\mathrm{m}(n,d)}$ containing $\{\mathscr{G}_0\}$. Then, by a version of the *Separating Hyperplane theorem* [47, Theorem 11.6, p. 100], we may conclude that $\mathscr{G}_0 \notin \mathrm{int}(\mathcal{G}_{\mathrm{m}(n,d)})$, which is a contradiction. Thus, $\mathscr{G}_0 \in \mathcal{G}_{\mathrm{sm}(n,d)}$, and therefore it follows that $\mathcal{G}_{\mathrm{sm}(n,d)} \supseteq \mathrm{int}(\mathcal{G}_{\mathrm{m}(n,d)})$.

Using $\partial$ to denote the boundary of a set, we conclude that, $\mathcal{G}_{\mathrm{m}(n,d)} \setminus \mathcal{G}_{\mathrm{sm}(n,d)} \subset \partial(\mathcal{G}_{\mathrm{m}(n,d)})$. Moreover as established before, $\mathcal{G}_{\mathrm{m}(n,d)}$ is a basic semialgebraic set. Thus, by [61, Theorem 1.8, p. 67], it follows that

$$\dim(\mathcal{G}_{\mathrm{m}(n,d)} \setminus \mathcal{G}_{\mathrm{sm}(n,d)}) \leq \dim\big(\partial(\mathcal{G}_{\mathrm{m}(n,d)})\big) < \dim(\mathcal{G}_{\mathrm{m}(n,d)}), \tag{A46}$$

which, since $\mathcal{G}_{\mathrm{m}(n,d)}$ is $\mu$-measurable, allows us to conclude that $\mu\!\restriction_{\mathcal{G}_{\mathrm{m}(n,d)}} (\mathcal{G}_{\mathrm{m}(n,d)} \setminus \mathcal{G}_{\mathrm{sm}(n,d)}) = 0$.

$\square$

### C.4 Proof of Theorem 4.3

**Theorem 4.3.** *For all $d \geq 2$, the set of SOS-monotone games of degree $d$ over a compact basic semialgebraic set $\mathcal{X}$ is dense in the set of monotone games of degree $d$ over $\mathcal{X}$. Furthermore, given any polynomial game $\mathscr{G}^* \in \mathcal{G}_{(n,d)}$ over $\mathcal{X}$, and any fixed $\ell \geq 0$, we can compute the closest $\ell$-SOS-monotone game to $\mathscr{G}^*$ by the program*

$$\begin{aligned} \underset{\mathscr{G} \in \mathcal{G}_{(n,d)}}{\text{minimize}} \quad & \|\mathscr{G} - \mathscr{G}^*\| \\ \text{subject to} \quad & \mathscr{G} \in \mathcal{G}_{\mathrm{sosm}(n,d,\ell)}, \end{aligned} \tag{19}$$

*which can be formulated as an SDP.*

*Proof.* Let $\mathcal{G}_{\mathrm{m}(n,d)}$ denote the set of $n$-player, $d$-degree polynomial monotone games. First, we show that $\mathcal{G}_{\mathrm{sosm}(n,d)}$ is dense in $\mathcal{G}_{\mathrm{m}(n,d)}$, i.e.,

$$\mathrm{cl}_{\mathcal{G}_{\mathrm{m}(n,d)}} \mathcal{G}_{\mathrm{sosm}(n,d)} \equiv \mathcal{G}_{\mathrm{m}(n,d)}. \tag{A47}$$

Let $\mathscr{G} \in \mathcal{G}_{\mathrm{m}(n,d)}$. Furthermore, for each $k \in \mathbb{N}$, define $\mathscr{G}_k$ as the $n$-player, $d$-degree polynomial game over $\mathcal{X}$ with utility functions $u_{k,1}, \dots, u_{k,n} \colon \mathcal{X} \to \mathbb{R}$ given by

$$u_{k,i}(x) = u_i(x) - \frac{1}{2k}\|x_i\|^2, \qquad \forall x \in \mathcal{X}, \qquad i \in [\![n]\!]. \tag{A48}$$

Then, the pseudo-gradient $v_k$ of $\mathscr{G}_k$ is given by

$$v_k(x) = \begin{pmatrix} \nabla_{x_1} u_{k,1}(x) \\ \vdots \\ \nabla_{x_n} u_{k,n}(x) \end{pmatrix} = \begin{pmatrix} \nabla_{x_1} u_1(x) - \frac{1}{k} \cdot x_1 \\ \vdots \\ \nabla_{x_n} u_n(x) - \frac{1}{k} \cdot x_n \end{pmatrix}, \qquad \forall x \in \mathcal{X}, \tag{A49}$$

and the symmetrized Jacobian matrix $\mathbf{SJ}_k$ of $v_k$ is given by

$$\mathbf{SJ}_k(x) = \mathbf{SJ}(x) - \frac{1}{k} \cdot \mathbf{I}, \qquad \forall x \in \mathcal{X}. \tag{A50}$$

Moreover, since $\mathscr{G}$ is monotone, it follows

$$\max_{x \in \mathcal{X}} \lambda_{\max}\big(\mathbf{SJ}_k(x)\big) = \max_{x \in \mathcal{X}} \lambda_{\max}\big(\mathbf{SJ}(x) - \frac{1}{k} \cdot \mathbf{I}\big) = \max_{x \in \mathcal{X}} \lambda_{\max}\big(\mathbf{SJ}(x)\big) - \frac{1}{k} \le -\frac{1}{k} < 0, \quad \text{(A51)}$$

and therefore $\mathscr{G}_k$ is strictly monotone. Subsequently, it follows that $\mathscr{G}_k$ is SOS-monotone, i.e., $\mathscr{G}_k \in \mathcal{G}_{\mathrm{sosm}(n,d)}$.

Now, consider the sequence $(\mathscr{G}_k)_{k=1}^{\infty}$ of SOS-monotone games. Observe that

$$\lim_{k \to \infty} \|\mathscr{G} - \mathscr{G}_k\| = \lim_{k \to \infty} \max_{i \in [\![n]\!]} \|\mathrm{vec}(u_i) - u(k,i)\|_{\infty} = \lim_{k \to \infty} \frac{1}{2k} = 0. \tag{A52}$$

In other words, $(\mathscr{G}_k)_{k=1}^{\infty}$ converges to $\mathscr{G}$. Thus, by definition, $\mathrm{cl}_{\mathcal{G}_{\mathrm{m}(n,d)}} \mathcal{G}_{\mathrm{sosm}(n,d)} \equiv \mathcal{G}_{\mathrm{m}(n,d)}$, i.e., $\mathcal{G}_{\mathrm{sosm}(n,d)}$ is dense in $\mathcal{G}_{\mathrm{m}(n,d)}$.

Next, we show that the program in (19) may be formulated as a SDP and solved in $\mathcal{O}\Big(\log\big(\frac{1}{\epsilon}\big) \cdot \mathrm{Poly}(\ell^2)\Big)$, where $\mathscr{G}^*([\![n]\!], \mathcal{X}, u^*) \in \mathcal{G}_{(n,d)}$, and $\ell \ge 0$.

First, by Definition 4.1, the program in (19) is equivalent to

$$\begin{aligned} \underset{\mathscr{G} \in \mathcal{G}_{(n,d)}}{\text{minimize}} \quad & \|\mathscr{G} - \mathscr{G}^*\| \\ \text{subject to} \quad & -y^{\mathsf{T}} \mathbf{SJ}(x)y \in Q_{\ell}(\mathcal{X} \times \mathcal{B}), \end{aligned} \tag{A53}$$

where $\mathbf{SJ}(x)$ is the (symmetrized) Jacobian matrix of the pseudo-gradient of $\mathscr{G}([\![n]\!], \mathcal{X}, u)$. Define the map $f \colon \big(\mathrm{vec}(u_1), \dots, \mathrm{vec}(u_n), x, y\big) \mapsto -y^{\mathsf{T}} \mathbf{SJ}(x)y$. Now, observe that $\big(\mathrm{vec}(u_1), \dots, \mathrm{vec}(u_n)\big) \mapsto \mathbf{SJ}(x)$ is an *affine map*, and therefore $f$ is affine in $\big(\mathrm{vec}(u_1), \dots, \mathrm{vec}(u_n)\big)$, and polynomial in $(x, y)$.

Next, by the definition of $\|\cdot\|$, we have that

$$\|\mathscr{G} - \mathscr{G}^*\| = \max_{i \in [\![n]\!]} \|\mathrm{vec}(u_i) - \mathrm{vec}(u_i^*)\|_{\infty}. \tag{A54}$$

Thus, the program (A53) is equivalent to

$$\begin{aligned} \underset{\lambda, \mathrm{vec}(u_1), \dots, \mathrm{vec}(u_n)}{\text{minimize}} \quad & \lambda \\ \text{subject to} \quad & \lambda \ge \max_{i \in [\![n]\!]} \|\mathrm{vec}(u_i) - \mathrm{vec}(u_i^*)\|_{\infty}, \\ & f\big(\mathrm{vec}(u_1), \dots, \mathrm{vec}(u_n), x, y\big) \in Q_{\ell}(\mathcal{X} \times \mathcal{B}) \end{aligned} \tag{A55}$$

Moreover, the condition $\lambda \ge \max_{i \in [\![n]\!]} \|\mathrm{vec}(u_i) - \mathrm{vec}(u_i^*)\|_{\infty}$ is equivalent to

$$\lambda \ge \mathrm{vec}(u_i)_j - \mathrm{vec}(u_i^*)_j, \qquad \forall i \in [\![n]\!], \, j \in \left[\!\!\left[ \binom{m_i + d}{d} \right]\!\!\right], \tag{A56a}$$

$$\lambda \ge \mathrm{vec}(u_i^*)_j - \mathrm{vec}(u_i)_j, \qquad \forall i \in [\![n]\!], \, j \in \left[\!\!\left[ \binom{m_i + d}{d} \right]\!\!\right]. \tag{A56b}$$

Thus, the program in (A55) is also equivalent to

$$\underset{\lambda, \text{vec}(u_1), \ldots, \text{vec}(u_n)}{\text{minimize}} \quad \lambda$$

$$\text{subject to} \quad \lambda \geq \text{vec}(u_i)_j - \text{vec}(u_i^*)_j, \quad \forall i \in [\![n]\!], \ j \in \left[\!\left[\binom{m_i + d}{d}\right]\!\right],$$

$$\lambda \geq \text{vec}(u_i^*)_j - \text{vec}(u_i)_j, \quad \forall i \in [\![n]\!], \ j \in \left[\!\left[\binom{m_i + d}{d}\right]\!\right]. \tag{A57}$$

$$f\big(\text{vec}(u_1), \ldots, \text{vec}(u_n), x, y\big) \in Q_\ell(\mathcal{X} \times \mathcal{B})$$

Finally, observe that, as $f$ is affine in $\text{vec}(u_1)$, $\ldots$, $\text{vec}(u_n)$, all the constraints of the program in (A57) are affine in $\lambda$, $\text{vec}(u_1)$, $\ldots$, $\text{vec}(u_n)$. Then, by definition, the program in (A57) is a SOS minimization program, and therefore it may be reformulated as a SDP. $\qquad\square$

## D  Extensive-Form Games with Imperfect Recall

### D.1  EFG Preliminaries

For completeness, we provide the necessary notation for EFGs with imperfect recall and their connection to polynomial optimization. First, we note that in standard game theory, strategies for EFGs lie in the simplex.

**Definition D.1.** An $n$-player extensive form game $\Gamma$ is a tuple $\Gamma := \langle \mathcal{H}, \mathcal{A}, \mathcal{Z}, u, \mathcal{I} \rangle$ where:

- The set $\mathcal{H}$ denotes the states of the game which are decision points for the players. The states $h \in \mathcal{H}$ form a tree rooted at an initial state $r \in \mathcal{H}$.

- Each state $h \in \mathcal{H}$ is associated with a set of *available actions* $\mathcal{A}(h)$.

- The set $\mathcal{N} := \{1, \ldots, n, c\}$ denotes the set of players of the game. Each state $h \in \mathcal{H}$ admits a label $\text{Label}(h) \in \mathcal{N}$ which denotes the *acting player* at state $h$. The letter $c$ denotes a special player called a *chance player*. Each state $h \in \mathcal{H}$ with $\text{Label}(h) = c$ is additionally associated with a function $\sigma_h : \mathcal{A}(h) \mapsto [0, 1]$ where $\sigma_h(a)$ denotes the probability that the chance player selects action $a \in \mathcal{A}(h)$ at state $h$, $\sum_{a \in \mathcal{A}(h)} \sigma_h(a) = 1$.

- $\text{Next}(a, h)$ denotes the state $h' := \text{Next}(a, h)$ which is reached when player $i := \text{Label}(h)$ takes action $a \in \mathcal{A}(h)$ at state $h$. $\mathcal{H}_i \subseteq \mathcal{H}$ denotes the states $h \in \mathcal{H}$ with $\text{Label}(h) = i$.

- $\mathcal{Z}$ denotes the terminal states of the game corresponding to the leafs of the tree. At each $z \in \mathcal{Z}$ no further action can be chosen, so $\mathcal{A}(z) = \varnothing$ for all $z \in \mathcal{Z}$. Each terminal state $z \in \mathcal{Z}$ is associated with value $u(z)$, where $u : \mathcal{Z} \to \mathbb{R}$ is called the utility function of the game.

- The game states $\mathcal{H}$ are further partitioned into *information sets* ascribed to each player, namely $\mathcal{I}_i \in (\mathcal{I}_1, \ldots, \mathcal{I}_n)$. Each information set $I \in \mathcal{I}_i$ encodes groups of nodes that the acting player $i$ cannot distinguish between, and thus the available actions within each infoset must be the same. Moreover, the player must play the same strategy in all nodes of the infoset. Formally, if $h_1, h_2 \in I$, then $\mathcal{A}(h_1) = \mathcal{A}(h_2)$. With slight abuse of notation, we can consider $\mathcal{A}(I)$ to be the set of shared available actions for the player in infoset $I$.

- For notational convenience, we ascribe a singleton information set to each chance node and define $\mathcal{I}_c$ as the collection of these chance node information sets. For each non-terminal node $h \in \mathcal{H} \notin \mathcal{Z}$, we thus define $I_h \in (\mathcal{I}_1, \ldots, \mathcal{I}_n) \cup \mathcal{I}_c$ to be its infoset.

The standard assumption in the literature is that of *perfect recall*, wherein no player ever forgets their past history (i.e. their past information sets and actions taken in those information sets) or any information acquired. Formally, for any infoset $I \in \mathcal{I}_i$ and for any two nodes $h_1, h_2 \in I$, the sequence of Player $i$'s actions from $r$ to $h_1$ and from $r$ to $h_2$ must coincide, otherwise they would be able to distinguish between the nodes. Finally, the game is called perfect recall if all players have perfect recall. Otherwise, the game is said to have the imperfect recall property. The notion of perfect recall has been crucial to establishing convergence results to pure Nash equilibria in extensive-form games, primarily via the concept of *behavioral strategies*:

**Definition D.2** (Behavioral Strategy). Consider the infosets belonging to player $i$, denoted $I \in \mathcal{I}_i$. Let $\Delta(\mathcal{A}(I))$ denote the set of probability distributions on the simplex over actions in $\mathcal{A}(I)$. The set of behavioral strategies of a player is denoted by $\sigma_i : \mathcal{I}_i \to \cup_{I \in \mathcal{I}_i} \Delta(\mathcal{A}(I))$. In particular, at each of their infosets $I$, player $i$ selects a probability distribution over their available actions at the infoset, $\sigma_i(\cdot|I) \in \Delta(\mathcal{A}(I))$. Finally, the joint behavioral strategy for all players is denoted $\sigma := (\sigma_i)_{i \in \mathcal{N}}$.

Similarly, mixed strategies can be defined in the following way:

**Definition D.3** (Mixed Strategy in Extensive-Form Games). Denote by $S_i$ the set of all possible actions across all game states $\mathcal{H}$ for player $i$ in $\Gamma$. Then, for all pure actions in the game $s \in S_i$, Player $i$'s mixed strategy $\mu_i$ is given by the probability distribution defined by the probabilities $\mu_i(s)$ of playing strategy $s$.

Intuitively, one can view behavioral strategies as players randomizing between their possible actions between each information set, and mixed strategies as players randomizing over their strategy sequences prior to playing the game (i.e. ex ante). Kuhn's theorem provides a meaningful connection between behavioral strategies and mixed strategies in EFGs with perfect recall:

**Theorem D.4** (Kuhn's Theorem [33]). *If player $i$ in an extensive form game has perfect recall, then for any mixed strategy $\mu$ of player $i$ there exists an equivalent behavioral strategy $\sigma$ of player $i$.*

Moreover, computing behavioral strategies in two-player zero-sum games of perfect recall is possible in polynomial time [31]. However, once the assumption of perfect recall is relaxed (i.e. when players have imperfect recall), Kuhn's theorem no longer holds and finding a solution even in the two-player zero-sum case becomes NP-complete [31].

## D.2 Imperfect Recall Games

[64] introduced an example of a game with no Nash equilibria in behavioral strategies (Figure 2). Subsequently, a variation of the original game called the forgetful penalty shoot-out game was introduced in [60] and proceeds as follows: Player 1 decides whether to kick a ball Left or Right before the whistle is blown, then decides again right before kicking the ball. At the second decision node, the player has forgotten their previous decision. If the decisions at the two nodes match, Player 1 manages to aim at the goal, during which Player 2 has to decide to dive Left or Right to stop the ball. Otherwise, the shot goes wide. This game also has no Nash equilibria in behavioral strategies.

When studying imperfect recall games, a key question to ask is whether one should consider mixed strategies or behavioral strategies. In particular, Kuhn's Theorem no longer holds and the convenient sequence form representation is not well-defined. Indeed, mixed strategies require players to select actions according to a distribution over all possible strategy sequences. For instance, a mixed strategy for Player 1 in the forgetful penalty shoot-out game (Figure 2) could look something like: Kick Left twice in a row with probability $0.5$, and kick Right twice in a row with probability $0.5$. However, this requires the players to have some memory of their previous actions. In contrast, behavioral strategies are more natural in imperfect recall games as they do not necessitate a memory requirement, a point which is argued for in Kuhn's original treatment of perfect recall games [33].

Following the work of [46, 59], we show a construction from imperfect recall EFGs to polynomial utilities via behavioral strategies. First, let $P(h'|\sigma, h)$ denote the realization probability of reaching $h'$ given that players using strategy $\sigma$ are at state $h$. Note that if $h \notin hist(h')$ (i.e., if $h'$ is not reachable from $h$) then the probability is $0$. Intuitively, the realization probability given a behavioral strategy is just the product of choice probabilities along the path from $h$ to $h'$. In order to formally define $P(h'|\sigma, h)$, we will need some additional notation. First, any node $h \in \mathcal{H}$ uniquely corresponds to a history $hist(h)$ from root $r$ to $h$.

- Function $\delta(h) : \mathcal{H} \to \mathbb{N}$ denotes the depth of the game tree starting from node $h \in \mathcal{H}$.

- Function $\nu(h, d) : \mathcal{H} \times \mathbb{N} \to \mathcal{H}$ identifies the node ancestor at depth $d \leq \delta$ from node $h$.

- Function $\alpha(h, d) : \mathcal{H} \times \mathbb{N} \to \cup_{h \in \mathcal{H}} \mathcal{A}(h)$ identifies the action ancestor at depth $d \leq \delta$ from node $h$.

Together, the sequence $(\nu(h, 0), \nu(h, 1), \ldots, \nu(h, \delta(h)))$ uniquely identifies the history of nodes from $r$ to $h$. Likewise, the sequence $(\alpha(h, 0), \alpha(h, 1), \ldots, \alpha(h, \delta(h) - 1))$ uniquely identifies the

history of actions taken from $r$ to $h$. Then, the realization probability of node $h'$ from $h$ if the players use joint strategy profile $\sigma$ is given by:

**Definition D.5** (Realization Probability).

$$P(h'|\sigma, h) = \prod_{j=\delta(h')}^{\delta(h)-1} \sigma(\alpha(h', j)|I_{\nu(h',j)}) \quad \text{if } h \in \text{hist}(h').$$

**Definition D.6** (Expected Utility for Player $i$). For player $i$ at node $h \in \mathcal{H} \setminus \mathcal{Z}$, if strategy profile $\sigma$ is played, their expected utility is given by $U_i(\sigma|h) := \sum_{z \in \mathcal{Z}} (P(z|\sigma, h) \cdot u_i(z))$. In its complete form, we can write the expected utility for each player as follows:

$$U_i(\sigma) = \sum_{z \in \mathcal{Z}} \left( \prod_{j=0}^{\delta(z)-1} \sigma(\alpha(z, j)|I_{\nu(z,j)}) \cdot u_i(z) \right)$$

With some abuse of notation, we can write $P(h|\sigma) := P(h|\sigma, r)$ where $r$ is the root node, and similarly $U_i(\sigma) := U_i(\sigma|r)$. Notice that by definition, the expected utility of each player is a polynomial function. In particular, $P(z|\sigma, h) \cdot u_i(z)$ is a monomial in $\sigma$ multiplied by a scalar.

Going forward, we establish several results connecting EFGs with imperfect recall and polynomial games, utilizing some additional notation.

- $\ell_i$ denotes the number of infosets of player $i$, i.e. $\ell_i := |\mathcal{I}_i|$. Moreover, fix an ordering $\left( I_i^1, \dots, I_i^{\ell_i} \right)$ of infosets in $\mathcal{I}_i$.

- $m_i^j$ denotes the number of actions in a given infoset $I_i^j \in \mathcal{I}_i$ of player $i$, i.e. $m_i^j := |\mathcal{A}(I_i^j)|$. Moreover, fix an ordering $\left( a_i^1, \dots, a_i^{\ell_i} \right)$ of actions in $\mathcal{A}(I_i^j)$.

- The strategy set of a player in information set $I$ is defined on the simplex $\Delta^{|\mathcal{A}(I)-1|}$, where $\Delta^{n-1} := \left\{ x \in \mathbb{R}^n : x_k \geq 0 \ \forall k, \sum_{k=1}^n x_k = 1 \right\}$.

- Subsequently, the strategy set of player $i$ over all of their infosets can be written as a Cartesian product of simplices: $\mathcal{S}_i := \bigtimes_{j=1}^{\ell_i} \Delta^{m_i^j - 1}$. Moreover, the strategy set over all players is $\mathcal{S} := \bigtimes_{i=1}^n \bigtimes_{j=1}^{\ell_i} \Delta^{m_i^j - 1}$.

- A joint strategy $\sigma \in \mathcal{S}$ for the players can hence be uniquely written as a vector $\sigma = (\sigma_{ik}^j)_{ijk} \in \bigtimes_{i=1}^n \bigtimes_{j=1}^{\ell_i} \Delta^{m_i^j - 1} \subset \bigtimes_{i=1}^n \bigtimes_{j=1}^{\ell_i} \mathbb{R}^{m_i^j}$.

Firstly, note that each infoset belonging to a player of an EFG with imperfect recall induces an additional variable in the expected utility function. Clearly, the resultant polynomial utilities can themselves be viewed as a polynomial game in the sense of [14, 45, 55] (and also falling in our definition of polynomial games $\mathcal{G}$), with the following definition of Nash equilibrium in behavioral strategies:

**Definition D.7** (Nash Equilibrium in Behavioral Strategies). A joint behavioral strategy $\sigma^* \in \bigtimes_{i=1}^n \bigtimes_{j=1}^{\ell_i} \Delta^{m_i^j - 1}$ is called a Nash equilibrium if for all players $i \in \mathcal{N}$:

$$U_i(\sigma^*) \geq U_i(\sigma_i, \sigma_{-i}^*), \quad \forall \sigma_i \in \mathcal{S}_i$$

i.e. no player has incentive to deviate from the behavioral strategy $\sigma^*$ in any of their information sets.

**Remark D.8.** The definition of Nash equilibria in our setting directly implies that any solution of the corresponding polynomial game defined using the polynomial utility functions is also a solution to the original EFG. In particular, the constructed polynomial utilities can be viewed as a generic polynomial game with utilities $u_i(x)$. Here, $x$ denotes the joint action profile of all players (see Section 2). Here, the number of variables in $x$ is equal to the total number of infosets over all players, $\sum_{i \in \mathcal{N}} \ell_i$. A joint state $x^*$ is called a Nash equilibrium if the following holds for all players $i \in \mathcal{N}$: $u_i(x^*) \geq u_i(x_i, x_{-i}^*) \forall x_i \in \mathcal{S}_i$. At a Nash equilibrium in the polynomial game, no player has incentive to unilaterally deviate in any of the variables they control. This is precisely the definition of Nash equilibrium in behavioral strategies for the original EFG.

## E   Additional Experimental Details

**Example 1: The absent-minded taxi driver in Figure 3.**   In the case of the game in Figure 3, we let $x$ denote the probability of choosing $C$ and $1 - x$ be the probability of choosing $E$. We use the SDP hierarchy in Eq. (17) to certify SOS-monotonicity of the polynomial $u(x) = -3x^2 + 4x$. We select $\ell = 2$ and obtain $\text{SOS}_2(\mathscr{G}) \approx -6 < 0$. Then, by Statement 4 of Theorem 3.2, the game is strictly monotone. This additionally guarantees that the solution of the game is unique [49].

**Example 2: A game with no Nash equilibria in Figure 2.**   Next, it follows as a consequence of Theorem 4.3 that we can use the program in Eq. (19) to find an SOS-monotone game $\mathscr{G}$ which is closest to the zero-sum game in Figure 2, in the sense of the norm defined in Eq. (3). By letting $x_1$ denote the probability that P1 selects $L$ at information set $\{a_1\}$, $x_2$ denote the probability that P1 selects $L$ at information set $\{a_2, a_3\}$, and $y$ denote the probability that P2 selects $r$, we obtain the payoff functions for P1 and P2 as follows:

$$u_1(x_1, x_2, y) = 10x_1x_2 + 2x_1y + 2x_2y - 6x_1 - 6x_2 - 2y + 1,$$

and $u_2 = -u_1$, respectively. Recall from [64] that this two-player zero-sum EFG does *not* have a NE and that the game is neither concave nor monotone. We first run our hierarchy of SOS optimization problems in Eq. (17) at level 2, and we attain an objective value of $\text{SOS}_2(\mathscr{G}) \approx 10 > 0$. Then, we run our program in Eq. (19) with additional constraints that $\mathscr{G}$ has to be zero-sum and that the information structure of the EFG has to be preserved. To preserve the information structure of the game, we select the monomial basis for the new payoff functions to be precisely the monomial basis that can appear in the original game. We obtain the closest SOS-monotone game $\mathscr{G}'$ given by:

$$u_1'(x_1, x_2, y) = -8x_1y - 8x_2y - 16x_1 - 16x_2 - 12y - 9,$$

and $u_2' = -u_1'$. The distance between the two games, $\|\mathscr{G} - \mathscr{G}'\|$, which is defined in Eq. (3), is in this case simply $\|\text{vec}(u_1) - \text{vec}(u_1')\|_\infty$ and equals 10. On the other hand, the payoff function of this game is multilinear and indeed the zero-sum game is monotone if and only if the term $x_1x_2$ has coefficient 0. This is in line with the experimental results. The modified game $\mathscr{G}'$ has zero symmetrized Jacobian matrix and is, thus, negative semidefinite. Moreover, since the modified game is SOS-monotone, it has a NE in behavioral strategies.

**Example 3.**   We generate a two-player polynomial game with the following payoff functions:

$$\begin{aligned}
u_1(x_1, x_2, y_1, y_2) = &- 0.5y_2^2 - 0.5y_1^2 - 0.5x_2^2 - 0.5x_1^2 - 9.365y_2^4 - 9.365y_1^2y_2^2 - 9.365y_1^4 \\
&- 1.171x_2^2y_2^2 + 0.08798x_2^2y_1y_2 - 0.9385x_2^2y_1^2 - 9.3654x_2^4 + 0.7825x_1x_2y_2^2 \\
&+ 0.5177x_1x_2y_1y_2 - 0.5465x_1x_2y_1^2 - 0.1310x_1^2y_2^2 - 0.1630x_1^2y_1y_2 \\
&- 0.1308x_1^2y_1^2 - 9.365x_1^2x_2^2 - 9.365x_1^4,
\end{aligned}$$

and

$$\begin{aligned}
u_2(x_1, x_2, y_1, y_2) = &- 0.5y_2^2 - 0.5y_1^2 - 0.5x_2^2 - 0.5x_1^2 - 6.828y_2^4 - 6.828y_1^2y_2^2 - 6.828y_1^4 \\
&- 0.8535x_2^2y_2^2 - 0.8631x_2^2y_1y_2 - 0.5324x_2^2y_1^2 - 6.828x_2^4 - 1.091x_1x_2y_2^2 \\
&- 1.699x_1x_2y_1y_2 - 0.4118x_1x_2y_1^2 - 0.3886x_1^2y_2^2 - 0.9771x_1^2y_1y_2 \\
&- 0.6141x_1^2y_1^2 - 6.828x_1^2x_2^2 - 6.828x_1^4.
\end{aligned}$$

P1 and P2 choose their actions $(x_1, x_2)$ and $(y_1, y_2)$ from a two-dimensional simplex respectively. This game is certified strictly monotone and SOS-monotone, as we run our hierarchy of SOS optimization problems in Eq. (17) and obtain an objective value $-1$ at level 4.

**Example 4.**   The game which was constructed is given in Figure E.4:

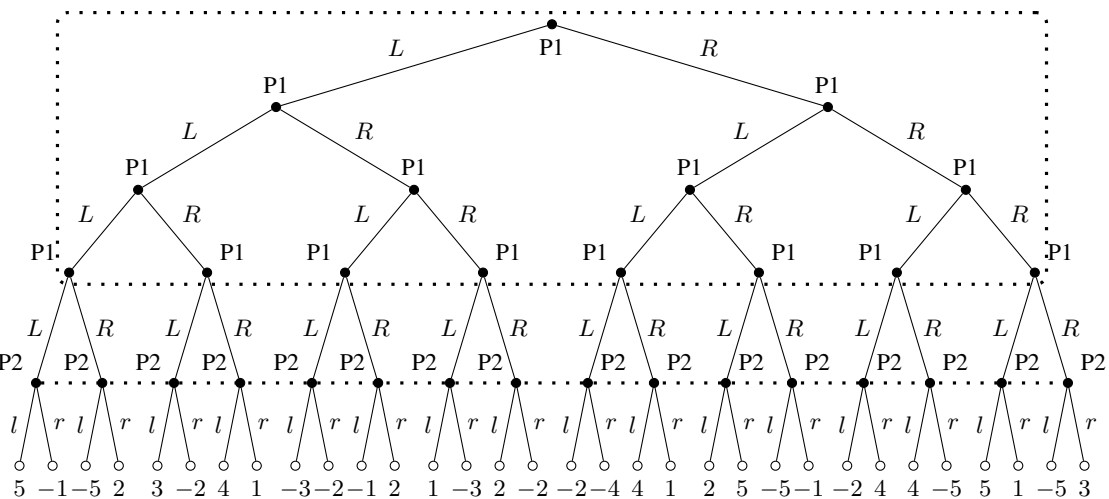

Figure E.4: Another Zero-sum Game with No Nash Equilibria

**Example 5.** We construct a two-player EFG with imperfect recall where P1 makes six moves before P2 makes two moves. The utility functions for both players are degree-8 polynomials with 4 variables, and we do not restrict the game to be zero-sum. The size of the monomial basis is 168, and we randomly generate the payoff functions for P1 and P2 by independently sampling the coefficient of each monomial in the basis from a uniform distribution on $[-1, 1]$. After rounding the coefficients to 2 s.f. for brevity, the payoff function for P1 is:

$$
\begin{aligned}
u_1(x, y) =\ & -0.42 - 0.46y_2 - 0.98y_1 + 0.93x_2 - 0.11x_1 - 0.78y_2^2 - 0.85y_1y_2 - 0.21y_1^2 + 0.92x_2y_2 + \\
& 0.48x_2y_1 + 0.29x_2^2 - 0.37x_1y_2 - 0.97x_1y_1 + 0.65x_1x_2 - 0.44x_1^2 - 0.86x_2y_2^2 - 0.35x_2y_1y_2 - \\
& 0.03x_2y_1^2 + 0.02x_2^2y_2 - 0.46x_2^2y_1 + 0.78x_2^3 - 0.87x_1y_2^2 - 0.59x_1y_1y_2 + 0.46x_1y_1^2 + 0.12x_1x_2y_2 + \\
& 0.37x_1x_2y_1 - 0.31x_1x_2^2 + 0.89x_1^2y_2 + 0.81x_1^2y_1 - 0.22x_1^2x_2 + 0.92x_1^3 + 0.85x_2^2y_2^2 + 0.34x_2^2y_1y_2 - \\
& 0.20x_2^2y_1^2 + 0.53x_2^3y_2 - 0.97x_2^3y_1 + 0.08x_2^4 + 0.98x_1x_2y_2^2 + 0.03x_1x_2y_1y_2 - 0.07x_1x_2y_1^2 + \\
& 0.10x_1x_2^2y_2 - 0.04x_1x_2^2y_1 - 0.33x_1x_2^3 + 0.41x_1^2y_2^2 + 0.61x_1^2y_1y_2 - 0.39x_1^2y_1^2 - 0.71x_1^2x_2y_2^2 + \\
& 0.84x_1^2x_2y_1 + 0.69x_1^2x_2^2 + 0.44x_1^3y_2 + 0.13x_1^3y_1 + 0.05x_1^3x_2 + 0.92x_1^4 - 0.10x_2^3y_2^2 - 0.55x_2^3y_1y_2 - \\
& 0.61x_2^3y_1^2 + 0.74x_2^4y_2 - 0.65x_2^4y_1 - 0.74x_2^5 - 0.14x_1x_2^2y_2^2 + 0.77x_1x_2^2y_1y_2 - 0.30x_1x_2^2y_1^2 + \\
& 0.41x_1x_2^3y_2 + 0.66x_1x_2^3y_1 - 0.62x_1x_2^4 - 0.39x_1^2x_2y_2^2 + 0.11x_1^2x_2y_1y_2 - 0.71x_1^2x_2y_1^2 - \\
& 0.14x_1^2x_2^2y_2 + 0.56x_1^2x_2^2y_1 + 0.60x_1^2x_2^3 + 0.26x_1^3y_2^2 + 0.34x_1^3y_1y_2 - 0.47x_1^3y_1^2 + 0.87x_1^3x_2y_2 + \\
& 0.29x_1^3x_2y_1 + 0.94x_1^3x_2^2 - 0.42x_1^4y_2 + 0.35x_1^4y_1 - 0.12x_1^4x_2 - 0.43x_1^5 + 0.04x_2^4y_2^2 + 0.15x_2^4y_1y_2 - \\
& 0.93x_2^4y_1^2 + 0.58x_2^5y_2 - 0.12x_2^5y_1 - 0.71x_2^6 + 0.93x_1x_2^3y_2^2 + 0.81x_1x_2^3y_1y_2 + 0.57x_1x_2^3y_1^2 + \\
& 0.32x_1x_2^4y_2 - 0.16x_1x_2^4y_1 - 0.46x_1x_2^5 + 0.85x_1^2x_2^2y_2^2 - 0.83x_1^2x_2^2y_1y_2 + 0.07x_1^2x_2^2y_1^2 - \\
& 0.60x_1^2x_2^3y_2 + 0.07x_1^2x_2^3y_1 + 0.44x_1^2x_2^4 - 0.14x_1^3x_2y_2^2 - 0.91x_1^3x_2y_1y_2 - 0.82x_1^3x_2y_1^2 + \\
& 0.52x_1^3x_2^2y_2 + 0.79x_1^3x_2^2y_1 + 0.39x_1^3x_2^3 - 0.04x_1^4y_2^2 - 0.61x_1^4y_1y_2 - 0.37x_1^4y_1^2 + 0.31x_1^4x_2y_2 - \\
& 0.85x_1^4x_2y_1 + 0.90x_1^4x_2^2 + 0.49x_1^5y_2 - 0.24x_1^5y_1 - 0.66x_1^5x_2 + 0.58x_1^6 - 0.57x_2^5y_2^2 + 0.72x_2^5y_1y_2 - \\
& 0.35x_2^5y_1^2 + 0.50x_2^6y_2 + 0.77x_2^6y_1 - 0.06x_1x_2^4y_2^2 + 0.89x_1x_2^4y_1y_2 - 0.48x_1x_2^4y_1^2 - 0.69x_1x_2^5y_2 + \\
& 0.61x_1x_2^5y_1 + 0.43x_1^2x_2^3y_2^2 + 0.16x_1^2x_2^3y_1y_2 - 0.58x_1^2x_2^3y_1^2 + 0.40x_1^2x_2^4y_2 - 0.34x_1^2x_2^4y_1 + \\
& 0.50x_1^3x_2^2y_2^2 + 0.20x_1^3x_2^2y_1y_2 - 0.77x_1^3x_2^2y_1^2 + 0.01x_1^3x_2^3y_2 + 0.05x_1^3x_2^3y_1 - 0.80x_1^4x_2y_2^2 - \\
& 0.04x_1^4x_2y_1y_2 + 0.26x_1^4x_2y_1^2 - 0.98x_1^4x_2^2y_2 + 0.91x_1^4x_2^2y_1 - 0.77x_1^5y_2^2 - 0.89x_1^5y_1y_2 - \\
& 0.72x_1^5y_1^2 + 0.17x_1^5x_2y_2 + 0.70x_1^5x_2y_1 + 0.81x_1^6y_2 - 0.57x_1^6y_1 + 0.31x_2^6y_2^2 + 0.82x_2^6y_1y_2 - \\
& 0.59x_2^6y_1^2 - 0.82x_1x_2^5y_2^2 - 0.72x_1x_2^5y_1y_2 + 0.93x_1x_2^5y_1^2 - 0.54x_1^2x_2^4y_2^2 + 0.66x_1^2x_2^4y_1y_2 + \\
& 0.69x_1^2x_2^4y_1^2 + 0.97x_1^3x_2^3y_2^2 + 0.28x_1^3x_2^3y_1y_2 + 0.32x_1^3x_2^3y_1^2 + 0.34x_1^4x_2^2y_2^2 - 0.82x_1^4x_2^2y_1y_2 + \\
& 0.49x_1^4x_2^2y_1^2 + 0.60x_1^5x_2y_2^2 - 0.95x_1^5x_2y_1y_2 + 0.14x_1^5x_2y_1^2 + 0.96x_1^6y_2^2 - 0.39x_1^6y_1y_2 - 0.28x_1^6y_1^2.
\end{aligned}
$$

Similarly, the payoff function for P2 is:

$$u_2(x,y) = -0.99 + 0.85y_2 + 0.77y_1 - 0.62x_2 - 0.88x_1 - 0.06y_2^2 - 0.32y_1y_2 - 0.57y_1^2 - 0.94x_2y_2 -$$
$$0.76x_2y_1 + 0.66x_2^2 - 0.11x_1y_2 - 0.32x_1y_1 - 0.53x_1x_2 + 0.47x_1^2 + 0.78x_2y_2^2 + 0.79x_2y_1y_2 -$$
$$0.98x_2y_1^2 + 0.65x_2^2y_2 - 0.58x_2^2y_1 + 0.01x_2^3 - 0.65x_1y_2^2 - 0.35x_1y_1y_2 + 0.95x_1y_1^2 - 0.86x_1x_2y_2 -$$
$$0.57x_1x_2y_1 + 0.76x_1x_2^2 + 0.64x_1^2y_2 + 0.28x_1^2y_1 + 0.86x_1^2x_2 - 0.74x_1^3 + 0.51x_2^2y_2^2 - 0.72x_2^2y_1y_2 -$$
$$0.41x_2^2y_1^2 + 0.39x_2^3y_2 - 0.70x_2^3y_1 + 0.37x_2^4 + 0.17x_1x_2y_2^2 - 0.12x_1x_2y_1y_2 - 0.43x_1x_2y_1^2 +$$
$$0.80x_1x_2^2y_2 + 0.34x_1x_2^2y_1 + 0.91x_1x_2^3 + 0.77x_1^2y_2^2 + 0.69x_1^2y_1y_2 + 0.64x_1^2y_1^2 + 0.84x_1^2x_2y_2 -$$
$$0.41x_1^2x_2y_1 - 0.01x_1^2x_2^2 - 0.46x_1^3y_2 + 0.94x_1^3y_1 - 0.33x_1^3x_2 + 0.65x_1^4 + 0.06x_2^3y_2^2 - 0.53x_2^3y_1y_2 -$$
$$0.81x_2^3y_1^2 + 0.44x_2^4y_2 + 0.32x_2^4y_1 + 0.74x_2^5 + 0.63x_1x_2^2y_2^2 + 0.96x_1x_2^2y_1y_2 - 0.21x_1x_2^2y_1^2 +$$
$$0.84x_1x_2^3y_2 + 0.13x_1x_2^3y_1 - 0.13x_1x_2^4 + 0.15x_1^2x_2y_2^2 - 0.46x_1^2x_2y_1y_2 - 0.90x_1^2x_2y_1^2 -$$
$$0.40x_1^2x_2^2y_2 - 0.07x_1^2x_2^2y_1 + 0.93x_1^2x_2^3 + 0.07x_1^3y_2^2 - 0.56x_1^3y_1y_2 + 0.33x_1^3y_1^2 + 0.08x_1^3x_2y_2 +$$
$$0.97x_1^3x_2y_1 + 0.86x_1^3x_2^2 - 0.50x_1^4y_2 - 0.44x_1^4y_1 + 0.59x_1^4x_2 + 0.98x_1^5 + 0.32x_2^4y_2^2 - 0.27x_2^4y_1y_2 +$$
$$0.81x_2^4y_1^2 - 0.39x_2^5y_2 - 0.47x_2^5y_1 - 0.23x_2^6 + 0.51x_1x_2^3y_2^2 + 0.64x_1x_2^3y_1y_2 + 0.25x_1x_2^3y_1^2 +$$
$$0.44x_1x_2^4y_2 + 0.89x_1x_2^4y_1 - 0.99x_1x_2^5 - 0.11x_1^2x_2^2y_2^2 + 0.14x_1^2x_2^2y_1y_2 + 0.98x_1^2x_2^2y_1^2 -$$
$$0.94x_1^2x_2^3y_2 + 0.53x_1^2x_2^3y_1 + 0.82x_1^2x_2^4 + 0.29x_1^3x_2y_2^2 - 0.68x_1^3x_2y_1y_2 + 0.28x_1^3x_2y_1^2 +$$
$$0.69x_1^3x_2^2y_2 + 0.08x_1^3x_2^2y_1 - 0.16x_1^3x_2^3 - 0.20x_1^4y_2^2 - 0.27x_1^4y_1y_2 - 0.23x_1^4y_1^2 + 0.62x_1^4x_2y_2 -$$
$$0.98x_1^4x_2y_1 - 0.35x_1^4x_2^2 + 0.39x_1^5y_2 + 0.33x_1^5y_1 - 0.59x_1^5x_2 - 0.23x_1^6 + 0.49x_2^5y_2^2 - 0.69x_2^5y_1y_2 +$$
$$0.93x_2^5y_1^2 + 0.54x_2^6y_2 + 0.38x_2^6y_1 - 0.28x_1x_2^4y_2^2 - 0.69x_1x_2^4y_1y_2 - 0.22x_1x_2^4y_1^2 - 0.32x_1x_2^5y_2 +$$
$$0.58x_1x_2^5y_1 + 0.60x_1^2x_2^3y_2^2 - 0.99x_1^2x_2^3y_1y_2 + 0.64x_1^2x_2^3y_1^2 + 0.69x_1^2x_2^4y_2 + 0.79x_1^2x_2^4y_1 +$$
$$0.45x_1^3x_2^2y_2^2 - 0.58x_1^3x_2^2y_1y_2 + 0.59x_1^3x_2^2y_1^2 + 0.39x_1^3x_2^3y_2 - 0.95x_1^3x_2^3y_1 + 0.68x_1^4x_2y_2^2 -$$
$$0.50x_1^4x_2y_1y_2 - 0.02x_1^4x_2y_1^2 + 0.60x_1^4x_2^2y_2 - 0.54x_1^4x_2^2y_1 - 0.80x_1^5y_2^2 - 0.22x_1^5y_1y_2 +$$
$$1.00x_1^5y_1^2 + 0.99x_1^5x_2y_2 + 0.82x_1^5x_2y_1 - 0.17x_1^6y_2 + 0.32x_1^6y_1 - 0.56x_2^6y_2^2 + 0.61x_2^6y_1y_2 +$$
$$0.98x_2^6y_1^2 + 0.76x_1x_2^5y_2^2 + 0.38x_1x_2^5y_1y_2 - 0.16x_1x_2^5y_1^2 - 0.16x_1^2x_2^4y_2^2 - 0.23x_1^2x_2^4y_1y_2 -$$
$$0.19x_1^2x_2^4y_1^2 + 0.43x_1^3x_2^3y_2^2 + 0.88x_1^3x_2^3y_1y_2 + 0.33x_1^3x_2^3y_1^2 + 0.09x_1^4x_2^2y_2^2 - 0.46x_1^4x_2^2y_1y_2 +$$
$$0.60x_1^4x_2^2y_1^2 + 0.17x_1^5x_2y_2^2 + 0.81x_1^5x_2y_1y_2 - 0.24x_1^5x_2y_1^2 + 0.05x_1^6y_2^2 - 0.72x_1^6y_1y_2 + 0.31x_1^6y_1^2.$$

We run our program in Eq. (19) at level 8 of the hierarchy to find the closest SOS-monotone game with an additional constraint that the information structure of the original EFG (i.e., the monomial basis) has to be preserved. This results in a modified game with the following payoff functions:

$$u_1'(x,y) = -0.92 - 0.96y_2 - 1.49y_1 + 0.42x_2 - 0.61x_1 - 1.28y_2^2 - 1.35y_1y_2 - 0.72y_1^2 + 1.03x_2y_2 +$$
$$0.76x_2y_1 - 0.21x_2^2 + 0.07x_1y_2 - 0.70x_1y_1 + 0.32x_1x_2 - 0.94x_1^2 - 0.41x_2y_2^2 - 0.33x_2y_1y_2 +$$
$$0.38x_2y_1^2 - 0.47x_2^2y_2 - 0.86x_2^2y_1 + 0.29x_2^3 - 0.39x_1y_2^2 - 0.29x_1y_1y_2 + 0.27x_1y_1^2 - 0.34x_1x_2y_2 -$$
$$0.05x_1x_2y_1 - 0.81x_1x_2^2 + 0.40x_1^2y_2 + 0.35x_1^2y_1 - 0.60x_1^2x_2 + 0.43x_1^3 + 0.35x_2^2y_2^2 + 0.19x_2^2y_1y_2 -$$
$$0.53x_2^2y_1^2 + 0.08x_2^3y_2 - 0.58x_2^3y_1 - 0.39x_2^4 + 0.51x_1x_2y_2^2 - 0.02x_1x_2y_1y_2 + 0.21x_1x_2y_1^2 -$$
$$0.12x_1x_2^2y_2 - 0.44x_1x_2^2y_1 - 0.14x_1x_2^3 - 0.08x_1^2y_2^2 + 0.24x_1^2y_1y_2 - 0.78x_1^2y_1^2 - 0.67x_1^2x_2y_2 +$$
$$0.44x_1^2x_2y_1 + 0.19x_1^2x_2^2 - 0.02x_2^3y_2 - 0.30x_2^3y_1 + 0.14x_2^3x_2 + 0.42x_1^4 - 0.46x_2^3y_2^2 - 0.49x_2^3y_1y_2 -$$
$$0.29x_2^3y_1^2 + 0.30x_2^4y_2 - 0.46x_2^4y_1 - 0.56x_2^5 - 0.43x_1x_2^2y_2^2 + 0.50x_1x_2^2y_1y_2 - 0.40x_1x_2^2y_1^2 +$$
$$0.17x_1x_2^3y_2 + 0.41x_1x_2^3y_1 - 0.20x_1x_2^4 - 0.11x_1^2x_2y_2^2 + 0.26x_1^2x_2y_1y_2 - 0.39x_1^2x_2y_1^2 -$$
$$0.49x_1^2x_2^2y_2 + 0.10x_1^2x_2^2y_1 + 0.14x_1^2x_2^3 - 0.07x_1^3y_2^2 + 0.30x_1^3y_1y_2 - 0.18x_1^3y_1^2 + 0.48x_1^3x_2y_2 -$$
$$0.07x_1^3x_2y_1 + 0.44x_1^3x_2^2 - 0.81x_1^4y_2 - 0.02x_1^4y_1 + 0.33x_1^4x_2 - 0.82x_1^5 - 0.23x_2^4y_2^2 - 0.17x_2^4y_1y_2 -$$
$$0.60x_2^4y_1^2 + 0.21x_2^5y_2 - 0.02x_2^5y_1 - 0.35x_2^6 + 0.55x_1x_2^3y_2^2 + 0.47x_1x_2^3y_1y_2 + 0.22x_1x_2^3y_1^2 +$$
$$0.14x_1x_2^4y_2 - 0.18x_1x_2^4y_1 - 0.03x_1x_2^5 + 0.41x_1^2x_2^2y_2^2 - 0.57x_1^2x_2^2y_1y_2 - 0.32x_1^2x_2^2y_1^2 -$$
$$0.59x_1^2x_2^3y_2 - 0.22x_1^2x_2^3y_1 - 0.01x_1^2x_2^4 - 0.33x_1^3x_2y_2^2 - 0.53x_1^3x_2y_1y_2 - 0.51x_1^3x_2y_1^2 +$$
$$0.10x_1^3x_2^2y_2 + 0.33x_1^3x_2^2y_1 - 0.07x_1^3x_2^3 - 0.43x_1^4y_2^2 - 0.22x_1^4y_1y_2 + 0.00x_1^4y_1^2 + 0.08x_1^4x_2y_2 -$$
$$0.62x_1^4x_2y_1 + 0.40x_1^4x_2^2 + 0.02x_1^5y_2 + 0.02x_1^5y_1 - 0.17x_1^5x_2 + 0.11x_1^6 - 0.17x_2^5y_2^2 + 0.34x_2^5y_1y_2 -$$
$$0.12x_2^5y_1^2 + 0.09x_2^6y_2 + 0.33x_2^6y_1 + 0.01x_1x_2^4y_2^2 + 0.50x_1x_2^4y_1y_2 - 0.21x_1x_2^4y_1^2 - 0.25x_1x_2^5y_2 +$$
$$0.22x_1x_2^5y_1 + 0.05x_1^2x_2^3y_2^2 + 0.05x_1^2x_2^3y_1y_2 - 0.43x_1^2x_2^3y_1^2 + 0.05x_1^2x_2^4y_2 - 0.22x_1^2x_2^4y_1 +$$
$$0.07x_1^3x_2^2y_2^2 - 0.02x_1^3x_2^2y_1y_2 - 0.61x_1^3x_2^2y_1^2 - 0.24x_1^3x_2^3y_2 - 0.07x_1^3x_2^3y_1 - 0.43x_1^4x_2y_2^2 +$$
$$0.04x_1^4x_2y_1y_2 + 0.06x_1^4x_2y_1^2 - 0.62x_1^4x_2^2y_2 + 0.44x_1^4x_2^2y_1 - 0.47x_1^5y_2^2 - 0.43x_1^5y_1y_2 -$$
$$0.28x_1^5y_2^2 - 0.01x_1^5x_2y_2 + 0.32x_1^5x_2y_1 + 0.35x_1^6y_2 - 0.12x_1^6y_1 + 0.04x_2^6y_2^2 + 0.41x_2^6y_1y_2 -$$
$$0.16x_2^6y_1^2 - 0.43x_1x_2^5y_2^2 - 0.30x_1x_2^5y_1y_2 + 0.51x_1x_2^5y_1^2 - 0.15x_1^2x_2^4y_2^2 + 0.27x_1^2x_2^4y_1y_2 +$$
$$0.28x_1^2x_2^4y_1^2 + 0.52x_1^3x_2^3y_2^2 - 0.08x_1^3x_2^3y_1y_2 + 0.15x_1^3x_2^3y_1^2 + 0.06x_1^4x_2^2y_2^2 - 0.40x_1^4x_2^2y_1y_2 +$$
$$0.12x_1^4x_2^2y_1^2 + 0.26x_1^5x_2y_2^2 - 0.53x_1^5x_2y_1y_2 + 0.09x_1^5x_2y_1^2 + 0.46x_1^6y_2^2 + 0.02x_1^6y_1y_2 + 0.10x_1^6y_1^2,$$

and

$$u_2'(x,y) = -1.5 + 0.34y_2 + 0.26y_1 - 1.12x_2 - 1.38x_1 - 0.55y_2^2 - 0.15y_1y_2 - 1.06y_1^2 - 0.83x_2y_2 -$$
$$0.48x_2y_1 + 0.16x_2^2 + 0.32x_1y_2 - 0.06x_1y_1 - 1.03x_1x_2 - 0.04x_1^2 + 0.30x_2y_2^2 + 0.65x_2y_1y_2 -$$
$$0.94x_2y_1^2 + 0.37x_2^2y_2 - 0.23x_2^2y_1 - 0.49x_2^3 - 0.40x_1y_2^2 - 0.05x_1y_1y_2 + 0.46x_1y_1^2 - 0.90x_1x_2y_2 -$$
$$0.60x_1x_2y_1 + 0.25x_1x_2^2 + 0.46x_1^2y_2 + 0.22x_1^2y_1 + 0.35x_1^2x_2 - 1.24x_1^3 + 0.03x_2^2y_2^2 - 0.45x_2^2y_1y_2 -$$
$$0.68x_2^2y_1^2 + 0.18x_2^3y_2 - 0.41x_2^3y_1 - 0.13x_2^4 - 0.21x_1x_2y_2^2 - 0.08x_1x_2y_1y_2 - 0.14x_1x_2y_1^2 +$$
$$0.70x_1x_2^2y_2 + 0.34x_1x_2^2y_1 + 0.41x_1x_2^3 + 0.32x_1^2y_2^2 + 0.50x_1^2y_1y_2 + 0.17x_1^2y_1^2 + 0.57x_1^2x_2y_2 -$$
$$0.38x_1^2x_2y_1 - 0.51x_1^2x_2^2 - 0.30x_1^3y_2 + 0.82x_1^3y_1 - 0.83x_1^3x_2 + 0.15x_1^4 - 0.41x_2^3y_2^2 - 0.23x_2^3y_1y_2 -$$
$$0.95x_2^3y_1^2 + 0.31x_2^4y_2 + 0.49x_2^4y_1 + 0.24x_2^5 + 0.27x_1x_2^2y_2^2 + 0.68x_1x_2^2y_1y_2 - 0.11x_1x_2^2y_1^2 +$$
$$0.58x_1x_2^3y_2 - 0.06x_1x_2^3y_1 - 0.63x_1x_2^4 - 0.15x_1^2x_2y_2^2 - 0.28x_1^2x_2y_1y_2 - 0.56x_1^2x_2y_1^2 -$$
$$0.34x_1^2x_2^2y_2 - 0.01x_1^2x_2^2y_1 + 0.42x_1^2x_2^3 - 0.30x_1^3y_2^2 - 0.40x_1^3y_1y_2 - 0.13x_1^3y_1^2 - 0.27x_1^3x_2y_2 +$$
$$0.85x_1^3x_2y_1 + 0.36x_1^3x_2^2 - 0.18x_1^4y_2 - 0.33x_1^4y_1 + 0.09x_1^4x_2 + 0.48x_1^5 - 0.12x_2^4y_2^2 - 0.06x_2^4y_1y_2 +$$
$$0.43x_2^4y_1^2 - 0.36x_2^5y_2 - 0.22x_2^5y_1 - 0.73x_2^6 + 0.11x_1x_2^3y_2^2 + 0.34x_1x_2^3y_1y_2 + 0.09x_1x_2^3y_1^2 +$$
$$0.26x_1x_2^4y_2 + 0.73x_1x_2^4y_1 - 1.50x_1x_2^5 - 0.33x_1^2x_2^2y_2^2 + 0.18x_1^2x_2^2y_1y_2 + 0.88x_1^2x_2^2y_1^2 -$$
$$0.84x_1^2x_2^3y_2 + 0.43x_1^2x_2^3y_1 + 0.32x_1^2x_2^4 - 0.01x_1^3x_2y_2^2 - 0.44x_1^3x_2y_1y_2 + 0.52x_1^3x_2y_1^2 +$$
$$0.46x_1^3x_2^2y_2 - 0.00x_1^3x_2^2y_1 - 0.66x_1^3x_2^3 + 0.00x_1^4y_2^2 + 0.01x_1^4y_1y_2 - 0.69x_1^4y_1^2 + 0.28x_1^4x_2y_2 -$$
$$0.78x_1^4x_2y_1 - 0.86x_1^4x_2^2 + 0.64x_1^5y_2 + 0.16x_1^5y_1 - 1.09x_1^5x_2 - 0.73x_1^6 + 0.05x_2^5y_2^2 - 0.39x_2^5y_1y_2 +$$
$$0.52x_2^5y_1^2 + 0.22x_2^6y_2 + 0.19x_2^6y_1 - 0.47x_1x_2^4y_2^2 - 0.85x_1x_2^4y_1y_2 - 0.06x_1x_2^4y_1^2 - 0.34x_1x_2^5y_2 +$$
$$0.26x_1x_2^5y_1 + 0.30x_1^2x_2^3y_2^2 - 0.82x_1^2x_2^3y_1y_2 + 0.56x_1^2x_2^3y_1^2 + 0.59x_1^2x_2^4y_2 + 0.67x_1^2x_2^4y_1 +$$
$$0.18x_1^3x_2^2y_2^2 - 0.51x_1^3x_2^2y_1y_2 + 0.51x_1^3x_2^2y_1^2 + 0.24x_1^3x_2^3y_2 - 0.77x_1^3x_2^3y_1 + 0.38x_1^4x_2y_2^2 -$$
$$0.31x_1^4x_2y_1y_2 + 0.23x_1^4x_2y_1^2 + 0.60x_1^4x_2^2y_2 - 0.47x_1^4x_2^2y_1 - 0.34x_1^5y_2^2 + 0.12x_1^5y_1y_2 +$$
$$0.52x_1^5y_2^2 + 0.58x_1^5x_2y_2 + 0.51x_1^5x_2y_1 + 0.11x_1^6y_2 + 0.16x_1^6y_1 - 0.34x_2^6y_2^2 + 0.19x_2^6y_1y_2 +$$
$$0.56x_2^6y_1^2 + 0.41x_1x_2^5y_2^2 + 0.10x_1x_2^5y_1y_2 - 0.38x_1x_2^5y_1^2 - 0.13x_1^2x_2^4y_2^2 - 0.21x_1^2x_2^4y_1y_2 -$$
$$0.04x_1^2x_2^4y_1^2 + 0.04x_1^3x_2^3y_2^2 + 0.55x_1^3x_2^3y_1y_2 + 0.05x_1^3x_2^3y_1^2 + 0.08x_1^4x_2^2y_2^2 - 0.18x_1^4x_2^2y_1y_2 +$$
$$0.23x_1^4x_2^2y_1^2 - 0.15x_1^5x_2y_2^2 + 0.60x_1^5x_2y_1y_2 - 0.03x_1^5x_2y_1^2 - 0.04x_1^6y_2^2 - 0.29x_1^6y_1y_2 - 0.14x_1^6y_1^2.$$

## F   Application: Economic Markets

Fisher markets are a special case of Arrow-Debreu markets [6] where competitive equilibria can be efficiently computed for specific classes of utility functions. In particular, Fisher markets are markets with $n$ buyers and $m$ divisible goods, and a certain amount of each good $j$ in the market, denoted $c_j > 0$. Each buyer $i$ comes to the market with a budget $w_i > 0$, and their objective is to obtain a bundle of goods $b_i \in \mathbb{R}_+^m$ that maximizes their utility function $u_i : \mathbb{R}_+^m \to \mathbb{R}_+$.

Computing competitive equilibria in Fisher markets is known to be PPAD-complete [10], but the works of Eisenberg and Gale [15, 16] showed that equilibrium computation is efficient if the buyers' utilities are continuous, concave and homogeneous. In recent years, many works have also leveraged techniques from algorithmic game theory to design algorithms that can compute competitive equilibria in Fisher markets in a decentralized fashion [12, 29, 13, 21, 24]. A majority of these prior works focus on Fisher markets where buyers' utilities are linear, quasilinear or Leontief.

In an effort to model more complex utility structures in markets, [22] initiated the first study on linear Fisher markets with a continuum of items. Subsequently, [65] introduced a variant of Fisher markets which captures the impact of social influence on buyers' utilities, showing that these markets can be viewed as *pseudo-games*, a construction from [6] which led directly to Rosen's definition of concave games. [11] also utilized a variational inequality approach to study monotone variants of these games, and presented decentralized algorithms that converge to equilibria. Indeed, SOS-concave and SOS-monotone games allow the study of Fisher markets with social influence for which concavity/monotonicity is verifiable. In particular, an economist who is constructing a model for a market can use our proposed methods in two ways. First, they can use the hierarchy in Eq. (17) to verify whether a game is concave or monotone. They can also use the hierarchy in Eq. (19) to search within the class of SOS-concave/monotone games in order to ensure that their market model satisfies equilibrium existence and even uniqueness.

