# OpenReview forum: "Certifying Concavity and Monotonicity in Games via Sum-of-Squares Hierarchies"
_NeurIPS.cc/2025/Conference — NeurIPS 2025 poster_

### Official Review · Reviewer_sBN1 · 2025-07-02

**Clarity:** 4
**Significance:** 4
**Originality:** 3
**Rating:** 5
**Confidence:** 4

**Summary:**

The work aims to certify concavity and monotonicity in games. First they demonstrate that certification is NP-hard for a class of games with contains imperfect recall EFGs. Then, they develop a SOS hierarchy for certification of concavity of monotonicity, that is polytime solvable. They demonstrate that most games fall into a finite level of this hierarchy, and show how to approximate general games one that falls into this hierarchy, and finally apply this to imperfect recall EFGs (which fall in the NP-hard class).

**Questions:**

What are some directions that build on top of this work? Is it primarily tackling the scalability of the proposed SDP formulations?

**Ethical Concerns:**

["NO or VERY MINOR ethics concerns only"]

**Final Justification:**

I maintain my assessment. The authors satisfactorily answered my question about future directions.

**Limitations:**

The authors address limitations.

**Quality:**

3

**Strengths And Weaknesses:**

The work is well-written and tackles an important problem, given that tractability of equilibrium computation is largely captured by concavity and monotonicity. Their story for the paper is coherent: they set out to solve certification, demonstrate that certification is not necessarily tractable in polytime (unless $\mathsf{P} = \mathsf{NP}$), develop a polytime computable SOS hierarchy, show that this is generic for monotone games, and then demonstrate a sequence of SDPs that correspond to the closest monotone games in levels of the hierarchy, and whose limit corresponds to the closest monotone game (under distances, including norms), and finally they apply this framework to EFGs with imperfect recall.

---

> ### Author Rebuttal · Authors · 2025-07-30
>
> We are grateful to the reviewer for their support of our paper. We will respond to your question below.
>
> > **Question 1: What are some future directions of our work?**
>
> We believe that scalability (as described in the responses to other reviewers) is one of the key future directions for this line of study. Another interesting direction is how to utilize the SOS framework for other applications. In EFGs, we are able to capture the information structure of the game (i.e. the monomial basis of the corresponding polynomial game) within the SDP, which allows us to find the closest SOS-concave/monotone game by changing the payoffs but not the game tree. A crucial question is how to perform similar analyses for other game theoretic settings while maintaining the appropriate structural properties.
>
> There are also several open theoretical questions. Characterizing the measure-zero set of non-strictly concave/monotone polynomial games would be helpful to understand the conditions under which our finite convergence guarantees fail. The proof of Thm 3.3 also relies on topological arguments tailored to polynomial games, and it would be interesting to see if similar results extend to other classes of continuous games. Finally, our techniques open a line of inquiry about the efficient verification of other desirable properties in games.

---

> > ### Comment · Reviewer_sBN1 · 2025-08-05
> >
> > Thank you for your response. I maintain my support of the paper as well as my score.

---

### Official Review · Reviewer_888D · 2025-07-03

**Clarity:** 3
**Significance:** 3
**Originality:** 3
**Rating:** 4
**Confidence:** 3

**Summary:**

The paper proposes hierarchies of sum-of-squares (SOS, a type of SDP) programs that certify concavity (which ensures the existence of Nash equilibria) and monotonicity (wihch ensures unicity) of an input game, where, for each level, the certification problem can be solved in polynomial time. It shows that most (i.e., excluding only a zero-measure set) concave/monotone games can be certified at some level of the hierarchy. It then shows how to approximate a given game with a SOS concave/monotone games in polytime. A case study on extensive-form games with imperfect recall is reported.

**Questions:**

- Can the authors comment a bit more of how their code can be used?

- Can the authors provide some information on the computing times of their certification/approximation algorithms, possibly w.r.t. the size of the input game and the level of the hierarchy the certification is taking place at? This would be very interesting.

- Regarding Thm 3.3: I just wonder whether some practically relevant games may not be captured due to belonging to the 0-measure set. Are there cases of practically relevant games which are isolated points in this space?

**Ethical Concerns:**

["NO or VERY MINOR ethics concerns only"]

**Final Justification:**

The authors' responses are satisfying.

**Limitations:**

Yes (none to report).

**Paper Formatting Concerns:**

All good.

**Quality:**

3

**Strengths And Weaknesses:**

With this said, I found the paper extremely well written and its results of clear interest. The appendices are also well written, and the last section (market application) is rather nice, albeit a little stylized. The authors indicated that their code is provided as a supplementary file, but didn't comment much (if at all) on this code in the paper, besides (I believe) using it in Sec. 6.1. I wonder if they could tell us a bit more of how their code can be used in practice. Also, no indication of the computing times is reported, besides a (reasonable) comment on the fact that SDPs are easy to solve in theory but not so much in practice). Some extra information on this would be very valuable.

---

> ### Author Rebuttal · Authors · 2025-07-30
>
> We are grateful to the reviewer for the thoughtful comments and questions, and for taking the time to carefully read our paper. We will respond to each of your concerns separately below.
>
> > **Comment 1: Better explanations of code in paper, and compute time for experiments.**
>
> We are happy to add more explanations of how the code can be used within the paper itself, including code snippets and accompanying descriptions for ease of use. To give an idea of how the computation time might scale, we have carried out three more experiments of larger scale examples, namely Examples 3–5. Example 3 is designed to be a strictly monotone game. Examples 4 and 5 are based on different canonical examples of EFGs with imperfect recall.
>
> **Example 3:** We generate a two-player polynomial game with the following payoff functions:
> $$
> \begin{align}
> u_1(x_1, x_2, y_1, y_2) &= -0.5 y_2^2 - 0.5 y_1^2 - 0.5 x_2^2 - 0.5 x_1^2 - 9.365 y_2^4 - 9.365 y_1^2 y_2^2 - 9.365 y_1^4 - 1.171 x_2^2 y_2^2 + 0.08798 x_2^2 y_1 y_2 - 0.9385 x_2^2*y_1^2 - 9.3654 x_2^4 \\\\ &\qquad + 0.7825 x_1 x_2 y_2^2 + 0.5177 x_1 x_2 y_1 y_2 - 0.5465 x_1 x_2 y_1^2 - 0.1310 x_1^2 y_2^2 - 0.1630 x_1^2 y_1 y_2 - 0.1308 x_1^2 y_1^2 - 9.365 x_1^2 x_2^2 - 9.365 x_1^4,
> \end{align}
> $$
> and
> 	$$
> \begin{align}
> u_2(x_1, x_2, y_1, y_2) &= -0.5 y_2^2 - 0.5 y_1^2 - 0.5 x_2^2 - 0.5 x_1^2 - 6.828 y_2^4 - 6.828 y_1^2 y_2^2 - 6.828 y_1^4 - 0.8535 x_2^2 y_2^2 - 0.8631 x_2^2 y_1 y_2 - 0.5324 x_2^2 y_1^2 - 6.828 x_2^4 \\\\ &\qquad - 1.091 x_1 x_2 y_2^2 - 1.699 x_1 x_2 y_1 y_2 - 0.4118 x_1 x_2 y_1^2 - 0.3886 x_1^2 y_2^2 - 0.9771 x_1^2 y_1 y_2 - 0.6141 x_1^2 y_1^2 - 6.828 x_1^2 x_2^2 - 6.828 x_1^4.
> \end{align}$$
> P1 and P2 choose their actions $(x_1, x_2)$ and $(y_1, y_2)$ from a two-dimensional simplex respectively. This game is certified strictly monotone and SOS-monotone, as we run our hierarchy of SOS optimization problems in Eq. (17) and obtain an objective value -1 at level 4. The SDP solving time at level 4 of the SOS hierarchy is 0.05225 seconds on an M3 MacBook Air with 16GB RAM.
>
> **Example 4:** While we are not allowed to add a figure to this response, we construct a two-player zero-sum EFG with imperfect recall which is a larger version of the game in Figure 2. In this game, P1 makes four moves before P2 makes one move. There is only one information set for P1 and one information set for P2. P1 has two actions to choose with probability $x$ and $1 - x$, respectively. P2 also has two actions to choose with probability $y$ and $1 - y$, respectively. Hence, the game tree has six layers including the root and the leaves (outcomes), and the payoff functions are degree-5 polynomials with monomial basis $[x^4 y, x^4, x^3 y, x^3, x^2 y, x^2, x y, x, y, 1]$. After we assign the payoff to each outcome in the game tree, the payoff function for P1 is obtained and given as follows:
> 	$$ u_1(x, y) = - 16 x^4 y + 25 x^4 + 74 x^3 y - 59 x^3 - 89 x^2 y + 49 x^2 + 45 x y - 19 x - 8 y + 3. $$
> A figure of the game tree will be added to future versions of the paper.
>
> We run our program in Eq. (19) at level 5 of the hierarchy to find the closest SOS-monotone game. Two additional constraints are imposed to retain the properties of the original EFG: a) the modified game has to be zero-sum, and b) the information structure of the original EFG has to be preserved. To preserve the information structure of the game, we select the monomial basis for the new payoff functions to be precisely that of the original game. The following modified payoff function is found, making the new EFG SOS-monotone:
> $$  u_1'(x, y) = - 5.6 x^4 y - 6 x^4 + 32.8 x^3 y - 22.9 x^3 - 75.1 x^2 y - 4 x y - 68 x - 57 y - 46. $$
> The SDP solving time for the program in Eq. (19) is 0.008975 seconds on the same MacBook Air.
>
> **Example 5:** We construct a two-player EFG with imperfect recall where P1 makes six moves before P2 makes two moves. The utility functions for both players are degree-8 polynomials with 4 variables, and we do not restrict the game to be zero-sum. The size of the monomial basis is 168, and we randomly generate the payoff functions for P1 and P2 by independently sampling the coefficient of each monomial in the basis from a uniform distribution on $[-1, 1]$. Due to limited space, we will include a full specification of the experiment in future versions of the paper.
>
> We run our program in Eq. (19) at level 8 of the hierarchy to find the closest SOS-monotone game with an additional constraint that the information structure of the original EFG has to be preserved. We will include the new utility functions in future versions of the paper. The SDP solving time for the program in Eq. (19) is 37.53 seconds, a significant increase from the smaller examples. This further motivates future work to improve scalability in large games (see our discussion of potential techniques in the response to Reviewer HE7B).
>
> In the camera ready version of the paper, we further aim to a) enhance the reusability, modularity and readability of our code to cover a broader range of games with different structures, b) perform new experiments on larger EFG examples with different game tree structures, and c) report code execution times for each experiment in a table, which will include the number of variables and constraints in the SDP.
>
>
> > **Question 1: On Theorem 3.3 and examples of concave/monotone games that belong on the measure-zero set.**
>
> Note that since all strict concave/monotone games have an SOS certificate at some finite level, these examples can only arise in non-strictly concave/monotone games. Topologically, these games lie on the boundary of the set of concave/monotone polynomial games. However, even small perturbations of the Hessians/Jacobian of a game can bring it into the interior, so in practice (and assuming approximation is allowed), one could perturb the game slightly to ensure strictness, thus guaranteeing finite convergence of the hierarchy.
>
> Nevertheless, there has been a line of work [1-4] studying learning in ‘merely monotone’ games, which are precisely games that are not-strictly monotone and thus cannot provably be certified at a finite level of our SOS hierarchy. The cited works contain several concrete examples, and notably the presence of coupling constraints (see e.g., [2] and references therein) on the players’ joint action set leads to non-strictly monotone utility functions. Clearly, these games are of interest, and it would be crucial to study certification methods with finite time guarantees for this class.
>
> *[1] Grammatico, Sergio. "Comments on “Distributed robust adaptive equilibrium computation for generalized convex games”[Automatica 63 (2016) 82–91]." Automatica 97 (2018): 186-188.*
>
> *[2] Tatarenko, Tatiana, and Maryam Kamgarpour. "Learning generalized Nash equilibria in a class of convex games." IEEE Transactions on Automatic Control 64.4 (2018): 1426-1439.*
>
> *[3] Tatarenko, Tatiana, and Maryam Kamgarpour. "Learning Nash equilibria in monotone games." 2019 IEEE 58th Conference on Decision and Control (CDC). IEEE, 2019.*
>
> *[4] Tatarenko, Tatiana, and Maryam Kamgarpour. "Bandit learning in convex non-strictly monotone games." arXiv preprint arXiv:2009.04258 (2020).*
>
> If the reviewer is satisfied with our response and their concerns have been addressed, we would greatly appreciate it if they would consider raising their score.

---

> > ### Comment · Reviewer_888D · 2025-08-05
> >
> > Dear authors,
> >
> > Thanks for your replies. I am mostly happy with them, albeit, I confess, what I was really hoping forwith my question on computing times was a more systematic analysis of the computing times to allow one to gauge the scalability of the proposed methods as the number of players/actions grows (in particular because SDPs are not as scalable as other convex-optimization problems).

---

> > > ### Author Response · Authors · 2025-08-06
> > > **Response to Follow-Up**
> > >
> > > Thank you for the feedback. We share your sentiment that a systematic analysis by varying the number of players/actions would be particularly useful to game theory practitioners. However, our work is primarily theoretical, and our experimental results are meant to be a proof-of-concept for game theorists, using only standard Julia SOS packages. With the many recent papers about improving SDP scalability in mind, we think it is crucial to benchmark modern methods in the literature (e.g. DSOS, SDSOS, low rank ADMM, HALLaR,…) in a systematic manner. This would require us to adjust/vary not just the game parameters but also the SDP solving technique and heuristics used. Thus, we believe that your suggestion is substantial enough to warrant a follow-up work. Concretely, to stay within the NeurIPS page limit, we will clean up and streamline our code as you have suggested, and add a table with the computing times for the experiments we have done (up to deg-8, 4 variables), which would hopefully give readers a high-level idea for what to expect in their own experiments.

---

> > > > ### Comment · Reviewer_888D · 2025-08-08
> > > >
> > > > Thanks for the reply (and for the honesty). I'm satisfied with the response.

---

### Official Review · Reviewer_HE7B · 2025-07-03

**Clarity:** 3
**Significance:** 3
**Originality:** 3
**Rating:** 5
**Confidence:** 2

**Summary:**

The paper begins by asking whether it is possible to efficiently verify concavity and monotonicity of polynomial games with semialgebraic strategy sets. To that end, it states and proves a theorem that verifying concavity and monotonicity of polynomial games of degree at least three is NP-hard and proceeds to identify a tractable hierarchy of increasingly strong sufficient conditions for concavity and monotonicity using the techniques of sum-of-squares optimization. Lastly, it applies this hierarchy to extensive-form games with imperfect recall.

**Questions:**

Theorem 3.3 states that **almost** all monotone games can be certified at a finite level. Do you have any examples or intuition for monotone games that cannot be certified at a finite level?

Do you have any preliminary results on the scalability of the approach to larger EFGs with imperfect recall?

**Ethical Concerns:**

["NO or VERY MINOR ethics concerns only"]

**Final Justification:**

I was already positive on the paper, and I would like to keep my score "accept"

**Limitations:**

Yes

**Quality:**

3

**Strengths And Weaknesses:**

Strengths:
The paper proves that verifying concavity and monotonicity is NP-hard which is a significant and important result and provides a possible solution to the intractability in the form of hierarchy of sufficient conditions.
The proved results apply to a general class of n-player polynomial games which include important class of extensive-form games with imperfect recall.
The experiments demonstrate the use of the theoretical framework on a small imperfect-recall game.

Weaknesses:
The preliminary section could provide a better introduction to sum-of-squares optimization for readers unfamiliar with the topic.
The experiments were limited to a single small imperfect recall game and given the scalability limitations of semidefinite programming, the applicability of the proposed approach to larger imperfect recall games remains unknown.
While the paper is well-written, rigorous and with a clear structure, given the complexity of the concepts discussed, it would benefit from a couple of illustrative examples.

---

> ### Author Rebuttal · Authors · 2025-07-30
>
> We are grateful to the reviewer for their support of our paper, and for the insightful feedback and questions. We will respond to each of your comments separately below.
>
> > **Comment 1: SOS preliminaries for unfamiliar readers.**
>
> We are happy to include a more comprehensive introduction to the SOS techniques in the preliminary section, which should hopefully make the paper self-contained for readers with a game-theory background.
>
> > **Comment 2: Regarding additional, larger experiments.**
>
> We have carried out three more experiments of larger scale examples, namely Examples 3–5. Example 3 is designed to be a strictly monotone game. Examples 4 and 5 are based on different canonical examples of EFGs with imperfect recall.
>
> **Example 3:** We generate a two-player polynomial game with the following payoff functions:
>
> $$
> \begin{align}
> u_1(x_1, x_2, y_1, y_2) &= -0.5 y_2^2 - 0.5 y_1^2 - 0.5 x_2^2 - 0.5 x_1^2 - 9.365 y_2^4 - 9.365 y_1^2 y_2^2 - 9.365 y_1^4 - 1.171 x_2^2 y_2^2 + 0.08798 x_2^2 y_1 y_2 - 0.9385 x_2^2*y_1^2 - 9.3654 x_2^4 \\\\ &\qquad + 0.7825 x_1 x_2 y_2^2 + 0.5177 x_1 x_2 y_1 y_2 - 0.5465 x_1 x_2 y_1^2 - 0.1310 x_1^2 y_2^2 - 0.1630 x_1^2 y_1 y_2 - 0.1308 x_1^2 y_1^2 - 9.365 x_1^2 x_2^2 - 9.365 x_1^4,
> \end{align}
> $$
> and
> 	$$
> \begin{align}
> u_2(x_1, x_2, y_1, y_2) &= -0.5 y_2^2 - 0.5 y_1^2 - 0.5 x_2^2 - 0.5 x_1^2 - 6.828 y_2^4 - 6.828 y_1^2 y_2^2 - 6.828 y_1^4 - 0.8535 x_2^2 y_2^2 - 0.8631 x_2^2 y_1 y_2 - 0.5324 x_2^2 y_1^2 - 6.828 x_2^4 \\\\ &\qquad - 1.091 x_1 x_2 y_2^2 - 1.699 x_1 x_2 y_1 y_2 - 0.4118 x_1 x_2 y_1^2 - 0.3886 x_1^2 y_2^2 - 0.9771 x_1^2 y_1 y_2 - 0.6141 x_1^2 y_1^2 - 6.828 x_1^2 x_2^2 - 6.828 x_1^4.
> \end{align}$$
> P1 and P2 choose their actions $(x_1, x_2)$ and $(y_1, y_2)$ from a two-dimensional simplex respectively. This game is certified strictly monotone and SOS-monotone, as we run our hierarchy of SOS optimization problems in Eq. (17) and obtain an objective value -1 at level 4. The SDP solving time at level 4 of the SOS hierarchy is 0.05225 seconds on an M3 MacBook Air with 16GB RAM.
>
> **Example 4:** While we are not allowed to add a figure to this response, we construct a two-player zero-sum EFG with imperfect recall which is a larger version of the game in Figure 2. In this game, P1 makes four moves before P2 makes one move. There is only one information set for P1 and one information set for P2. P1 has two actions to choose with probability $x$ and $1 - x$, respectively. P2 also has two actions to choose with probability $y$ and $1 - y$, respectively. Hence, the game tree has six layers including the root and the leaves (outcomes), and the payoff functions are degree-5 polynomials with monomial basis $[x^4 y, x^4, x^3 y, x^3, x^2 y, x^2, x y, x, y, 1]$. After we assign the payoff to each outcome in the game tree, the payoff function for P1 is obtained and given as follows:
> 	$$ u_1(x, y) = - 16 x^4 y + 25 x^4 + 74 x^3 y - 59 x^3 - 89 x^2 y + 49 x^2 + 45 x y - 19 x - 8 y + 3. $$
> A figure of the game tree will be added to future versions of the paper.
>
> We run our program in Eq. (19) at level 5 of the hierarchy to find the closest SOS-monotone game. Two additional constraints are imposed to retain the properties of the original EFG: a) the modified game has to be zero-sum, and b) the information structure of the original EFG has to be preserved. To preserve the information structure of the game, we select the monomial basis for the new payoff functions to be precisely that of the original game. The following modified payoff function is found, and the new EFG is SOS-monotone:
> $$  u_1'(x, y) = - 5.6 x^4 y - 6 x^4 + 32.8 x^3 y - 22.9 x^3 - 75.1 x^2 y - 4 x y - 68 x - 57 y - 46. $$
> The SDP solving time for the program in Eq. (19) is 0.008975 seconds on the same MacBook Air.
>
> **Example 5:** We construct a two-player EFG with imperfect recall where P1 makes six moves before P2 makes two moves. The utility functions for both players are degree-8 polynomials with 4 variables, and we do not restrict the game to be zero-sum. The size of the monomial basis is 168, and we randomly generate the payoff functions for P1 and P2 by independently sampling the coefficient of each monomial in the basis from a uniform distribution on $[-1, 1]$. Due to limited space, we will include a full specification of the experiment in future versions of the paper.
>
> We run our program in Eq. (19) at level 8 of the hierarchy to find the closest SOS-monotone game with an additional constraint that the information structure of the original EFG (i.e. the monomial basis) has to be preserved. We will include the new utility functions in future versions of the paper. The SDP solving time for the program in Eq. (19) is 37.53 seconds, a significant increase from the smaller examples. This further motivates potential future work on scalability (see below).
>
> In the camera ready version of the paper, we further aim to a) enhance the reusability, modularity and readability of our code to cover a broader range of games with different structures, b) perform new experiments on larger EFG examples with different game tree structures, and c) report code execution times for each experiment in a table, which will include the number of variables and constraints in the SDP.
>
> > **Comment 3: Regarding future work on scalability.**
>
> Improving scalability is a highly active subarea of research in semidefinite/polynomial programming, with many directions proposed in recent years. As mentioned in the paper, DSOS and SDSOS [1] trade off compute for solution quality in SOS optimization. Separately, recent work has utilized low-rank approaches to improve empirical runtimes for SDPs [2-5]. While these more empirical results are not the focus of the present work, we agree with the reviewers that applying these ideas in large-scale games (concretely, even imperfect recall variants of well-studied large EFGs like poker and bridge) would be exciting future work.
>
> *[1] Ahmadi, Amir Ali, and Anirudha Majumdar. "DSOS and SDSOS optimization: more tractable alternatives to sum of squares and semidefinite optimization." SIAM Journal on Applied Algebra and Geometry 3.2 (2019): 193-230.*
>
> *[2] Monteiro, Renato DC, Arnesh Sujanani, and Diego Cifuentes. "A low-rank augmented Lagrangian method for large-scale semidefinite programming based on a hybrid convex-nonconvex approach." arXiv preprint arXiv:2401.12490 (2024).*
>
> *[3] Han, Qiushi, et al. "A low-rank ADMM splitting approach for semidefinite programming." arXiv preprint arXiv:2403.09133 (2024).*
>
> *[4] Han, Qiushi, et al. "Accelerating low-rank factorization-based semidefinite programming algorithms on GPU." arXiv preprint arXiv:2407.15049 (2024).*
>
> *[5] Aguirre, Jacob M., et al. "cuHALLaR: A GPU Accelerated Low-Rank Augmented Lagrangian Method for Large-Scale Semidefinite Programming." arXiv preprint arXiv:2505.13719 (2025).*
>
>
> > **Question 1: On Theorem 3.3 and examples of concave/monotone games that cannot be certified at a finite level.**
>
> Note that since all strict concave/monotone games have an SOS certificate at some finite level, these examples can only arise in non-strictly concave/monotone games. Topologically, these games lie on the boundary of the set of concave/monotone polynomial games. However, even small perturbations of the Hessians/Jacobian of a game can bring it into the interior, so in practice (and assuming approximation is allowed), one could perturb the game slightly to ensure strictness, thus guaranteeing finite convergence of the hierarchy.
>
> There has been a line of work [6-9] studying learning in ‘merely monotone’ games, which are precisely games that are not-strictly monotone and thus cannot provably be certified at a finite level of our SOS hierarchy. The cited works contain several examples, and notably the presence of coupling constraints (see e.g., [7] and references therein) on the players’ joint action set leads to non-strictly monotone utility functions. Clearly, these games are of interest, and it would be crucial to study certification methods with finite time guarantees for this class.
>
> *[6] Grammatico, Sergio. "Comments on “Distributed robust adaptive equilibrium computation for generalized convex games”[Automatica 63 (2016) 82–91]." Automatica 97 (2018): 186-188.*
>
> *[7] Tatarenko, Tatiana, and Maryam Kamgarpour. "Learning generalized Nash equilibria in a class of convex games." IEEE Transactions on Automatic Control 64.4 (2018): 1426-1439.*
>
> *[8] Tatarenko, Tatiana, and Maryam Kamgarpour. "Learning Nash equilibria in monotone games." 2019 IEEE 58th Conference on Decision and Control (CDC). IEEE, 2019.*
>
> *[9] Tatarenko, Tatiana, and Maryam Kamgarpour. "Bandit learning in convex non-strictly monotone games." arXiv preprint arXiv:2009.04258 (2020).*

---

### Official Review · Reviewer_o8ht · 2025-07-09

**Clarity:** 2
**Significance:** 2
**Originality:** 2
**Rating:** 4
**Confidence:** 3

**Summary:**

This paper investigates the computational tractability of verifying two key structural properties of games: **concavity** and **monotonicity**. These properties are fundamental in algorithmic game theory as they enable the existence and efficient computation of Nash equilibria and underpin favorable convergence guarantees for learning algorithms. The authors establish that checking whether a game is concave or monotone is NP-hard, when the utilities are polynomial functions. Despite this negative result, the authors propose a hierarchy of semidefinite programming (SDP) relaxations (Theorem 3.2) based on the sum-of-squares (SOS) paradigm that can serve as certificates for concavity or monotonicity. They show that almost all (Theorem 3.3) concave or monotone games can be certified at some finite level of this hierarchy. Additionally, they define subclasses (Section 4) of SOS-concave and SOS-monotone games and show that one can efficiently compute the closest such game (in a certain norm) to any given polynomial game. Finally, some of techniques are applied to extensive-form games with imperfect recall.

**Questions:**

As I explained in the above section, this paper is mainly a theoretical contribution within the specific setting of polynomial programming and sum-of-squares methods. Because of this focus, many broader questions (for example, whether the authors have considered non-polynomial utility functions) are likely to fall outside the scope of the work and would not be particularly meaningful to ask.

Also, in line 66 “campact” should be “compact.”

**Ethical Concerns:**

["NO or VERY MINOR ethics concerns only"]

**Final Justification:**

Overall, I believe the paper presents a meaningful storyline and offers a clear and perspective on the problem. The reason I am maintaining my current score is that, in my opinion, a clear accept would require the introduction of genuinely new ideas and techniques. In contrast, the *majority* of the results in this work rely on known methods, with only minor (or at most moderate) adjustments.

**Limitations:**

Yes. The paper focuses on theoretical contributions and proof-of-concept experiments, and I did not identify significant unaddressed limitations or societal impacts.

**Paper Formatting Concerns:**

I did not notice any major formatting issue.

**Quality:**

3

**Strengths And Weaknesses:**

The paper is motivated by a clear and relevant question in game theory and optimization. Overall, the presentation is quite clear, and the authors include sufficient technical details to understand the approach. One concern is that parts of the results feel somewhat incremental. Given that the utilities are polynomial functions, the hardness of deciding concavity and monotonicity seems to follow rather directly from existing hardness results on polynomial nonnegativity and convexity. Similarly, the construction of the SDP hierarchy appears to be a natural application of known techniques from the sum-of-squares literature; the idea is simply to use the Rayleigh–Ritz theorem to express the problem of the largest eigenvalue as a maximization problem over a polynomial, which is a well-known result in linear algebra. Overall, the main result (Theorem 3.2) is mostly an adaptation of existing ideas to this setting and does not introduce fundamentally new techniques beyond this adaptation. That said, the work does a good job of tying these concepts together in the context of games, and the examples involving imperfect recall illustrate that the method can have practical implications in cases that are otherwise challenging to analyze.

---

> ### Author Rebuttal · Authors · 2025-07-30
>
> We are grateful to the reviewer for the thoughtful comments and questions, and for taking the time to carefully read our paper. We will respond to each of your concerns separately below.
>
> > **Comment 1: The results feel somewhat incremental, and do not introduce fundamentally new techniques.**
>
> Firstly, regarding our hardness result (Theorem 3.1), it builds upon recent work by Ahmadi et al. on the hardness of deciding polynomial convexity. However, their results do not cover monotonicity, and as such, appropriate modifications were required to handle this distinct property. Our goal with this result is not to claim novelty in the specific reduction technique, but to emphasize a foundational computational barrier for the game theory community: although concavity and monotonicity are highly desirable properties for games, determining whether a game satisfies them is computationally hard. Given the centrality of these properties to equilibrium analysis we believe it is crucial to make this hardness explicit before developing efficient certification methods.
>
> Secondly, while prior work on SOS-convexity (e.g., Ahmadi et al. 2013) focused on *unconstrained* settings, our setting involves *constrained* action sets, as is natural in games. Certifying concavity or monotonicity in such cases requires extending the SOS-convexity framework to constrained, semialgebraic domains. To our knowledge, there is no prior work that defines or analyzes SOS-based convexity or monotonicity in this setting, making our extension both necessary and novel.
>
> Thirdly, a key theoretical contribution of our paper is the result, proved using non-standard topological arguments, that for almost all monotone games, monotonicity can be certified at a finite level of the SOS hierarchy. This insight is central to our work: it opens a promising path toward tractable and certifiable analysis of games and positions our hierarchy not merely as an adaptation, but as a meaningful tool to navigate the geometric structure of game-theoretic solution spaces.
>
> Finally, our application of this framework to equilibrium computation/approximation in extensive-form games with imperfect recall constitutes a significant contribution in its own right. This class of games is notoriously challenging, with limited tools available for equilibrium computation. Our approach provides a novel and non-trivial bridge between optimization techniques and game-theoretic equilibrium analysis, potentially broadening the applicability of SOS methods in strategic settings.
>
> > **Question 1: Regarding continuous games with non-polynomial utilities.**
>
> We would like to reiterate that although the polynomial game setup may initially seem restrictive, it is in fact highly expressive and already supports significant applications, particularly due to its connection to extensive-form games, as discussed in the paper. Our proposed toolbox provides a new framework for equilibrium computation and game approximation in a class of problems that are both meaningful for applications and computationally challenging.
>
> While we do not explicitly address non-polynomial games in this work, we note that, by the Stone–Weierstrass approximation theorem [1], polynomials are universal approximators of continuous functions. As such, polynomial games form a natural and foundational subclass of continuous games and can serve as surrogates for approximating more general settings. Extending our toolbox to accommodate non-polynomial utilities, in conjunction with approximation results like Stone–Weierstrass, is a promising direction we are currently exploring.
>
>
> *[1] Stone, Marshall H. "The generalized Weierstrass approximation theorem." Mathematics Magazine 21.5 (1948): 237-254.*
>
>
> If the reviewer is satisfied with our response and their concerns have been addressed, we would greatly appreciate it if they would consider raising their score.

---

> ### Comment · Reviewer_o8ht · 2025-08-05
>
> Thank you to the authors for their detailed responses, and I apologize for the delay in my reply. I will address their comments in the order in which they were presented.
>
> > **Firstly, regarding our hardness result (Theorem 3.1), it builds upon recent work by Ahmadi et al. on the hardness of deciding polynomial convexity.**
>
> First, I would like to point out a small but important typo in the proof of Theorem 3.1: it is claimed that the gradient of a scalar function is monotone *iff* the function is concave. This is not correct. Instead, the statement is true for the gradient of a convex function. This can be easily corrected by replacing $p(x)$ with $-p(x)$.
>
> More substantively, I am not convinced by the authors' response to my original point. From my perspective, both hardness results concerning concavity and monotonicity follow immediately from Theorem 2.3 in Ahmadi et al. (2020); defining a game in which one player’s utility is a polynomial and the other’s is zero reduces the problem of verifying game concavity to verifying the concavity of the polynomial. Similarly, monotonicity follows from the above fundamental fact from convex analysis combined with Theorem 2.3 in Ahmadi et al. (2020). This appears to be a case of an easy application of a previous result.
>
> > **Secondly, while prior work on SOS-convexity (e.g., Ahmadi et al. 2013) focused on unconstrained settings, our setting involves constrained action sets, as is natural in games. […] To our knowledge, there is no prior work that defines or analyzes SOS-based convexity or monotonicity in this setting, making our extension both necessary and novel.**
>
> The authors emphasize that prior work on SOS-convexity, such as Ahmadi et al. (2013), considered unconstrained domains, while their paper focuses on constrained action sets (e.g., strategies on a simplex).
>
> However, I have several concerns. First, the construction closely mirrors the proof of Theorem 3.1, where instead of employing Theorem 2.3 in Ahmadi et al. (2020) they employ a similar result for polynomials defined on simplices.
>
> Additionally, Theorem 2.1 in Ahmadi et al. (2013) applies to unconstrained polynomials, not to functions defined on simplices, but in the proof they reference it. Moreover, the correct result is proven in Guo (1996), specifically for polynomials over simplices. Thus, simply by mirroring the above construction and using Guo (1996), they obtain Theorem C.2. My question is: why do the authors claim novelty? Once again, this appears to be an easy application of past results.
>
> > **Thirdly, a key theoretical contribution of our paper is the result, proved using non-standard topological arguments, that for almost all monotone games, monotonicity can be certified at a finite level of the SOS hierarchy.**
>
> There are a few technical details that I missed in my initial review. For example, in line 653, the authors write: “is the (uncountable) intersection of convex sets, and therefore it is convex.” However, this is only valid if the intersection is nonempty, which should be explicitly justified.
>
> In line 664, the phrase should read “containing ${\mathcal{G}_0}$” for clarity. Furthermore, I am uncertain about the validity of the following statement: “Thus, $L$ describes a non-trivial supporting hyperplane […]”. Is it not possible that for some **monotone** game $\mathcal{G}$, we have $L(\mathcal{G}) = 0$, given that $L$ is defined for a fixed $(x_0, u)$? Some elaboration on this would be helpful. Most probably I am missing something, so I would like to let the authors enlighten me on this; otherwise, their proof would need a correction.
>
> That said, I do find Theorem 3.3 valuable, particularly in conjunction with Item 4 of Theorem 3.2. However, I would like to clarify my questions.
>
> > **Extending our toolbox to accommodate non-polynomial utilities, in conjunction with approximation results like Stone–Weierstrass, is a promising direction we are currently exploring.**
>
> The authors suggest extending their framework to non-polynomial utilities using the Stone–Weierstrass theorem. While this is an intriguing direction, I am not entirely convinced of its feasibility. To put it simply, if one were to use it to approximate a continuous function $f$ with polynomials, then it would be natural to attempt constructing an SOS hierarchy to decide the non-negativity of an arbitrary $f$ (rather than a polynomial, as has been done in the past). This would likely have been the first step in your attempt to extend it within a game-theoretic framework to decide properties such as monotonicity and concavity, similar to what is done in this work. Even though the following is not a conclusive argument, the fact that this has not been done before—despite the long history of this literature—probably suggests that it is far from being feasible.

---

> > ### Comment · Reviewer_o8ht · 2025-08-06
> >
> > References
> >
> > - Ahmadi, A. A., Hall, G., & Parrilo, P. A. (2013). NP-hardness of deciding convexity of quartic polynomials and related problems.
> > - Ahmadi, A. A., Hall, G., & Parrilo, P. A. (2020). On the complexity of detecting convexity over a box.
> > - Guo, B. (1996). On the difficulty of deciding the convexity of polynomials over simplexes.
> >
> > I look forward to the authors’ response. While I still hold the concerns outlined above, I would be happy to engage in further discussion and clarification.

---

> > > ### Comment · Reviewer_o8ht · 2025-08-07
> > >
> > > I thank the authors for the valuable discussion we had and for addressing my concerns and comments. I would like to believe that the discussion was also valuable from the authors’ side.
> > >
> > > Although some of the results are derived through applications of existing work, the overall contribution is satisfying. Starting from the NP-hardness results for deciding the relevant properties, and then moving to the construction of the SOS hierarchy for certifying them, the paper presents a coherent line of reasoning. As I mentioned earlier, an interesting point comes from a topological argument—namely, that almost all monotone games are strictly monotone. This, together with Item 4 in the Theorem 3.2, shows that almost all monotone games can be certified at a finite level of the hierarchy $^{[1]}$. Of course, a natural question arises: is there an upper bound on this level or if one could distinguish classes of games where an upper bound on the level of the hierarchy can be shown? As I note in the footnote below, a similar question may have been raised in the context of certifying the non-negativity of polynomials.
> > >
> > > $^{[1]}$ I believe a similar phenomenon likely holds for certifying the non-negativity of polynomials. Put simply: for almost all non-negative polynomials, those that are not strictly positive form a measure-zero set.

---

> > > > ### Author Response · Authors · 2025-08-08
> > > > **Response to Reviewer o8ht**
> > > >
> > > > Thank you for your valuable comments during the rebuttal process, they have helped us improve our paper significantly and we greatly appreciate your efforts.
> > > >
> > > > To answer your question, with some additional assumptions there exists an upper bound on the level of the hierarchy which is doubly exponential (see Theorem 6 of [1]), for the Putinar-type certificates that we seek. At a high level, the parameters that affect the bound are: the description of the semialgebraic action set $A$, the degree of the polynomial $f$, and a measure of how close $f$ is to having a zero on $A$. Moreover, related to the second part of your question, recent work has shown that the exponential dependence can be improved in restricted cases, such as over the hypercube [2]. Then, the level required is bounded by $O(f_{max}/f_{min})$, where $f_{max}$ and $f_{min}$ are the max and min of the polynomial over the hypercube, respectively. We will certainly explore this avenue in future work, as this would allow for '’effective’' versions of our SOS results that specify, given a class of games, upper bounds on the hierarchy level required for certification.
> > > >
> > > > Regarding your footnote, this is actually a very interesting and natural direction -- we have tried but could not locate such a result in the literature. Hence, it is possible that we could extend our result to the general polynomial optimization case, which would be a useful contribution for the general SOS community as well.
> > > >
> > > > [1] Nie, J., & Schweighofer, M. (2007). On the complexity of Putinar's Positivstellensatz. Journal of Complexity, 23(1), 135-150.
> > > >
> > > > [2] Baldi, L., & Slot, L. (2024). Degree bounds for Putinar’s Positivstellensatz on the hypercube. SIAM Journal on Applied Algebra and Geometry, 8(1), 1-25.

---

> ### Author Response · Authors · 2025-08-06
> **Response to Follow-Up**
>
> Thank you for the careful reading of our work and the follow-up comments. We will respond to your statements separately:
>
> > **First, I would like to point out a small but important typo in the proof of Theorem 3.1...**
>
> Thank you for finding the typo, we will correct it in the next version of the paper.
>
> > **More substantively, I am not convinced by the authors' response to my original point. From my perspective, both hardness results concerning concavity and monotonicity follow immediately from Theorem 2.3 in Ahmadi et al. (2020). They do not introduce novel techniques but are instead straightforward applications of known results.**
>
> We would like to apologize for some unclear writing in our initial rebuttal, which might have caused some confusion.
>
> In our rebuttal, we explicitly mention that ''our goal with this result (i.e. Theorem 3.1) is *not* to claim novelty in the specific reduction technique, but to emphasize a foundational computational barrier for the game theory community''. Indeed, we fully agree with the Reviewer that the proofs of Thms 3.1 and C.2 follow almost immediately from past results. We hope this clarifies this point of our rebuttal: the hardness results are not technically novel, but are given as theorems due to their importance to the research question in a game theory context. We are happy to make this point clearer in a future version of the paper. Also, if the reviewer deems necessary, we are happy to remove the theorem environment.
>
> Our second paragraph (“Secondly, while…”) was meant to refer to Theorem 3.2 (i.e. the main SOS hierarchy result). Here, we also agree with the reviewer that the high-level techniques used are standard in the SOS literature. The purpose of this result is to summarize the canonical properties typically expected when applying the sum-of-squares framework, and we provide the first derivation of these properties in the setting of concave/monotone games. This type of high-level theorem is common in papers that employ the SOS technique, and is intended to highlight key properties for readers who may not be deeply familiar with the literature. As before, we are happy to make it clear in the manuscript that this result arises from standard techniques, even though they are applied in a novel context.
>
> > **(Regarding Thm 3.3) There are a few technical details that I missed in my initial review...**
>
> Thank you for your technical comments, please find our clarifications to your points below:
> 1. Note that the argument on Line 653 remains valid even if the intersection was empty because the empty set is convex. We will clarify this in the proof going forward.
> 2. Line 664 indeed has a typo, it should read ''containing $\mathcal{G}_0$''. Thank you for spotting it!
> 3. For the last argument, recall that we want to reach a contradiction by showing that if $\mathcal{G}_0$ is a strictly monotone game, it cannot lie in the relative interior of the set of monotone games. To show this we construct a supporting hyperplane that contains $\mathcal{G}_0$. In particular the function $L$ satisfies that $L(\mathcal{G}) \geq 0$ for all monotone games $\mathcal{G}$ (since $x_0 \in X$). Furthermore, $L(\mathcal{G}_0) = 0$ by construction. The last statement (that $L \not\equiv 0$) follows because of the existence of games that are strictly monotone for the entire space. We do not claim that $L(\mathcal{G}) > 0$ for all strictly monotone games, only that there exists one such game. This is the only condition required to invoke Rockafellar’s Theorem. We apologize for the clumsy writing in this part, and will improve it in the next version.
>
> > **if one were to use Stone–Weierstrass to approximate a continuous function f with polynomials, then it would be natural to attempt to construct an SOS hierarchy to decide properties such as non-negativity for an arbitrary f (instead of a polynomial as it has been done in the past)**
>
> As our first point, we emphasize that games with polynomial utilities are the sole focus of the present paper. As we explain in the manuscript, they capture a broad and practically significant class of applications, extensive-form games with imperfect recall, which are a central focus of our work.
>
> Regarding the non-polynomial case, we note that it lies outside the scope of this paper. That said, one common approach in the literature is to approximate (upper/lower bound) certain classes of non-polynomial functions via Taylor/Pade approximations; see, for example, Section 3.1 in [1]. This is a promising direction that we intend to explore in future research, though it is not addressed in the present study.
>
> [1] Faust, O., & Fawzi, H. (2023). Sum-of-squares proofs of logarithmic Sobolev inequalities on finite markov chains. IEEE Transactions on Information Theory, 70(2), 803-819.

---

### Note · Authors · 2025-08-13

We thank all reviewers for their time and invaluable feedback. During the discussion period, we have committed to making the following changes to our paper:

1. **Scalability and Experiments:** We have performed multiple new experiments on larger scale EFGs with different structures based on canonical EFG examples. This constitutes a first step towards addressing the main concerns of Reviewers HE7B and 888D regarding the scalability of our approach. We will also refactor our code to allow easier usage out-of-the-box, and add a table summarizing the experimental results and compute times to the main text.

2. **Technical Novelty and Proofs:** We will clarify technical details regarding the proof techniques used in Theorems 3.1, 3.2 and 3.3 within the text. We will also fix some typos and unclear writing which were brought up by Reviewer o8ht.

3. **SOS Preliminaries:** We will refactor the current SOS preliminaries section to be more digestible to a game-theory/ML community.

These changes complement our work’s contributions, which include: showing the *intractability of verifying concavity and monotonicity* in polynomial games, introducing an *SOS-based hierarchy* that can verify concavity/monotonicity in finite time, constructing an *SDP which can find the closest SOS-concave/monotone game* to a given game efficiently, and applying our proposed techniques to *EFGs of imperfect recall*.

Finally, there have been many interesting directions which arose during the review process, including: a) extending SOS techniques to games with non-polynomial utilities, b) improving scalability using modern methods that speed-up SDP-solving, and c) characterizing the measure-zero set of non-strictly concave/monotone games and for which our techniques do not have a finite-time guarantee.

The reviewers' feedback has significantly strengthened our paper, and we are grateful that Reviewers o8ht and 888D have found our responses satisfactory. Thank you for your consideration.

---

### Decision · Program_Chairs · 2025-09-17

**Decision:**

Accept (poster)

**Comment:**

This paper investigates the computational complexity of verifying concavity and monotonicity in games. The authors prove that verifying concavity and monotonicity of polynomial games of degree at least three is NP-hard, and identify a tractable hierarchy of increasingly strong sufficient conditions for concavity and monotonicity using the techniques of sum-of-squares optimization.  They also apply this hierarchy to extensive-form games with imperfect recall.  The reviewers find the results meaningful, and the presentation clear.  It is noted that some results appear to build on existing ideas, but the way these concepts are organized appears insightful.  There's also room for improvement concerning readability by people unfamiliar with the topic.  We encourage the authors to take the constructive comments into consideration and further improve the paper.